# RETHINKING THE SMOOTHNESS OF NODE FEATURES LEARNED BY GRAPH CONVOLUTIONAL NETWORKS

## ABSTRACT

It has been proved that graph convolutional layers (GCLs) using ReLU or leaky ReLU activation function smooth node features. Such a smoothing process is beneficial for node classification using a few GCLs. However, deep graph convolutional networks (GCNs) tend to learn homogeneous node feature vectors over the graph, making nodes indistinguishable. In this paper, we develop a new understanding of the smoothness of node features learned by GCNs by establishing a fine-grained analysis of how ReLU or leaky ReLU affects the smoothness of its input vectors. First, we establish a geometric relationship between the input and output of ReLU or leaky ReLU. Then we show that if one ignores the magnitude of the feature vectors, ReLU and leaky ReLU smooth their input feature vectors, echoing existing theory. We further show that taking the magnitude of feature vectors into account, ReLU and leaky ReLU can increase, decrease, or preserve the smoothness of their input vectors. Our theory informs the design of a simple yet effective approach to let GCN learn node features with a desired smoothness that improves its empirical performance for graph node classification.

## 1 INTRODUCTION

Let $G = (V, E)$ be an undirected graph where $V = \{v_i\}_{i=1}^n$ is the set of nodes and $E$ is the set of edges. Let $\boldsymbol{A} \in \mathbb{R}^{n \times n}$ be the adjacency matrix of the graph $G$ with $A_{ij} = \mathbf{1}_{(i,j) \in E}$, where $\mathbf{1}$ is the indicator function. Also, let $\boldsymbol{G}$ be the (augmented) normalized adjacency matrix, given by

$$\boldsymbol{G} := (\boldsymbol{D} + \boldsymbol{I})^{-\frac{1}{2}} (\boldsymbol{I} + \boldsymbol{A})(\boldsymbol{D} + \boldsymbol{I})^{-\frac{1}{2}} = \tilde{\boldsymbol{D}}^{-\frac{1}{2}} \tilde{\boldsymbol{A}} \tilde{\boldsymbol{D}}^{-\frac{1}{2}}, \tag{1}$$

where $\boldsymbol{I}$ is the identity matrix, $\boldsymbol{D}$ is diagonal with $D_{ii} = \sum_{j=1}^n A_{ij}$, and $\tilde{\boldsymbol{A}} := \boldsymbol{A} + \boldsymbol{I}$ and $\tilde{\boldsymbol{D}} := \boldsymbol{D} + \boldsymbol{I}$ are the adjacency matrix and degree matrix augmented with self-loops, respectively. Starting from the initial features $\boldsymbol{H}^0 := [(\boldsymbol{h}_1^0)^\top, \ldots, (\boldsymbol{h}_n^0)^\top]^\top \in \mathbb{R}^{d \times n}$ with $\boldsymbol{h}_i^0$ being the $i^{th}$ node feature, the graph convolutional layer (GCL) [22] learns node representations using the following transformation:

$$\boldsymbol{H}^l = \sigma(\boldsymbol{W}^l \boldsymbol{H}^{l-1} \boldsymbol{G}), \tag{2}$$

where $\sigma$ is the activation function, and $\boldsymbol{W}^l \in \mathbb{R}^{d \times d}$ is learnable. GCL smooths feature vectors of the neighboring nodes. The smoothness of node features has been argued to help node classification; see e.g. [24; 33; 6]; resonating with the idea of classical semi-supervised learning approaches [43; 40]. Besides graph convolutional networks (GCNs) that stack GCLs, many other graph neural networks (GNNs) have been developed based on different mechanisms, including spectral methods [4; 11], spatial methods [14; 32], sampling methods [15; 38], and the attention mechanism [32]. Many other GNN models can be found in recent surveys or monographs; see, e.g. [17; 2; 35; 41; 16].

Deep neural networks can have better predictive performance than shallow ones; an example is convolutional neural networks [23; 18]. However, such a property does not hold for GCNs; it has been observed that deep GCNs that consist of multiple GCLs perform significantly worse than shallow models with only a few GCLs [6]. In particular, the feature vectors learned by deep GCNs tend to be identical to each other over each connected component of the graph; this phenomenon is referred to as over-smoothing; see e.g. [24; 27; 28; 5; 6; 34], which not only occurs for GCN but also for many other GNNs, e.g. GraphSage [15] and MPNN [14]. Intuitively, each GCL smooths neighboring node features, which benefits node classification [24; 33; 6]; however, simply stacking these smoothing layers will inevitably make node features homogeneous. Algorithms have been developed to alleviate the over-smoothing of GNNs, including decoupling prediction and message passing [13], skip connection and batch normalization [20; 9; 7], graph sparsification [31], jumping

knowledge [36], scattering transform [26], PairNorm normalization layer [39], and controlling the Dirichlet energy of node features [42]. Besides over-smoothing, the authors of [1] suggest that over-squashing causes the underperformance of deep GCNs.

From a theoretical perspective, [28; 5] prove that deep GCNs using ReLU or leaky ReLU tend to make features indistinguishable. In particular, [28] shows that the distance of node features to the eigenspace $\mathcal{M}$ – corresponding to eigenvalue 1 of the matrix $G$ in equation (1) – goes to zero when the depth of GCN with ReLU activation goes to infinity. Moreover, [5] proves that the Dirichlet energy of the node features goes to zero when the depth of GCN with ReLU or leaky ReLU activation goes to infinity. A crucial step in the proofs in [28; 5] is that ReLU and leaky ReLU reduces the distance of feature vectors to $\mathcal{M}$ and the Dirichlet energy of feature vectors. However, [5] points out that ***over-smoothing – measured by the distance of node features to the eigenspace $\mathcal{M}$ or the Dirichlet energy – is a misnomer***, and the real smoothness of a graph signal should be characterized by a ***normalized smoothness***, e.g., normalizing the Dirichlet energy by the magnitude of the features. Nevertheless, analyzing the normalized smoothness of node features learned by GCN with ReLU or leaky ReLU activation remains an open problem. Moreover, it is interesting to ask if analyzing the normalized smoothness can bring in any new understanding of features learned by GCN.

## 1.1 OUR CONTRIBUTION

This paper aims to reexamine how each GCL with ReLU or leaky ReLU activation function smooths node features using both normalized and unnormalized smoothness notions, i.e. the smoothness of feature vectors both with and without considering their magnitude. In particular, we focus on studying how ReLU and leaky ReLU in GCNs affect the smoothness of node features. Based on our theoretical study, we also design an effective algorithm that can control the smoothness of learned node features in GCNs. We summarize our main contributions – especially theoretical results to ease the reading of our paper – as follows:

- We prove that the projection of the output of ReLU/leaky ReLU onto the eigenspace $\mathcal{M}^{\perp}$ – corresponding to eigenvalue 1 of matrix $G$ in equation (1) – lies in a high-dimensional sphere, whose center only depends on the input but the radius depends on both input and output of ReLU/leaky ReLU. This geometric characterization not only implies theories in [28; 5] but also informs that adjusting the projection of input onto the eigenspace $\mathcal{M}$ can effectively change the smoothness of the output. See Section 3 for details.
- We show that both ReLU and leaky ReLU reduce the distance of node features to the eigenspace $\mathcal{M}$, i.e. ReLU and leaky ReLU smoothness their input without considering the magnitude of feature vectors. In contrast, we show that when considering the magnitude of feature vectors, ReLU and leaky ReLU can increase, decrease, or preserve the smoothness of each dimension of their input vectors; see Sections 3 and 4.
- Inspired by our established geometric relationship between the input and output of ReLU or leaky ReLU, we study how adjusting the projection of input onto the eigenspace $\mathcal{M}$ affects the smoothness of the output. We show that the distance of the output to the eigenspace $\mathcal{M}$ is always no greater than that of the original input – no matter how we adjust the input by changing its projection onto $\mathcal{M}$. In contrast, adjusting the projection of input onto $\mathcal{M}$ can effectively change the normalized smoothness of output to any desired value. The details can be found in Section 4.
- Based on our theory, we propose a learnable smoothness control term (SCT) to let GCN and related networks adjust the projection of input onto the eigenspace $\mathcal{M}$ automatically. The resulting feature vectors learned by GCNs have a desired smoothness that can empirically improve the node classification accuracy. We comprehensively validate the benefits of our proposed SCT in improving node classification – for both homophilic and heterophilic graphs – using a few of the most representative GCN-style architectures. See Sections 5 and 6 for more details.

As far as we know, our work is the first thorough study of how ReLU and leaky ReLU affect the smoothness of node features both with and without considering their magnitude. Moreover, our theory informs a new practical algorithm to improve GCN for graph node classification.

## 1.2 ADDITIONAL RELATED WORKS

Controlling the smoothness of node features to improve the performance of GCNs is another line of related works. For instance, [39] designs a normalization layer to prevent node features from becoming too similar to each other, and the authors of [42] constrain the Dirichlet energy to control the smoothness of node features without a complete consideration of the activation function. While there has been effort on understanding and alleviating the over-smoothing of GCNs and controlling the

smoothness of node features, there is a shortage of theoretical examination of how activation functions affect the smoothness of node features, specifically accounting for the magnitude of features.

## 1.3 NOTATION

We denote scalars by lower- or upper-case letters and vectors and matrices by lower- and upper-case boldface letters, respectively. We denote the $\ell_2$-norm of a vector $\boldsymbol{u}$ as $\|\boldsymbol{u}\|$. For vectors $\boldsymbol{u}$ and $\boldsymbol{v}$, we use $\langle \boldsymbol{u}, \boldsymbol{v} \rangle$, $\boldsymbol{u} \odot \boldsymbol{v}$, and $\boldsymbol{u} \otimes \boldsymbol{v}$ to denote their inner, Hadamard, and Kronecker product, respectively; see Appendix A for details. For a matrix $\boldsymbol{A}$, we denote its $(i, j)^{th}$ entry, transpose, and inverse as $A_{ij}$, $\boldsymbol{A}^\top$, and $\boldsymbol{A}^{-1}$, respectively. We denote the trace of $\boldsymbol{A} \in \mathbb{R}^{n \times n}$ as $\text{Trace}(\boldsymbol{A}) = \sum_{i=1}^{n} A_{ii}$. For two matrices $\boldsymbol{A}$ and $\boldsymbol{B}$, we denote the Frobenius inner product as $\langle \boldsymbol{A}, \boldsymbol{B} \rangle_F := \text{Trace}(\boldsymbol{A}\boldsymbol{B}^\top)$ and the Frobenius norm of $\boldsymbol{A}$ as $\|\boldsymbol{A}\|_F := \sqrt{\langle \boldsymbol{A}, \boldsymbol{A} \rangle}$.

## 1.4 ORGANIZATION

We provide preliminaries and review existing results in Section 2. In Section 3, we establish a geometric characterization of how ReLU and leaky ReLU affect the smoothness of their input vectors. We study the smoothness of each dimension of node features and take their magnitude into account in Section 4. Our proposed SCT is presented in Section 5. We comprehensively verify the efficacy of the proposed SCT for improving node classification using three most representative GCN-style models in Section 6. Technical proofs and more experimental results are provided in the appendix.

## 2 PRELIMINARIES AND EXISTING RESULTS

From the spectral graph theory [10], we know that the eigenvalues of $\boldsymbol{G}$ in equation (1) can be sorted in descending order $1 = \lambda_1 = \ldots = \lambda_m > \lambda_{m+1} \geq \ldots \geq \lambda_n > -1$, where $m$ is the number of connected components of the graph $G$, i.e. we can decompose the vertex set $V = \{v_k\}_{k=1}^{n}$ into $m$ connected components $V_1, \ldots, V_m$. Let $\boldsymbol{u}_i = (\mathbf{1}_{\{v_k \in V_i\}})_{1 \leq k \leq n}$ be the indicator vector of the $i^{th}$ component $V_i$, i.e. the $k^{th}$ coordinate of $\boldsymbol{u}_i$ is one if the $k^{th}$ node $v_k$ lies in the connected component $V_i$; otherwise, is zero. Moreover, let $\boldsymbol{e}_i$ be the eigenvector associated with $\lambda_i$, then $\{\boldsymbol{e}_i\}_{i=1}^{n}$ forms an orthonormal basis of $\mathbb{R}^n$. Notice that $\{\boldsymbol{e}_i\}_{i=1}^{m}$ spans the eigenspace $\mathcal{M}$ – corresponding to eigenvalue 1 of matrix $\boldsymbol{G}$, and $\{\boldsymbol{e}_i\}_{i=m+1}^{n}$ spans the orthogonal complement of $\mathcal{M}$, denoted by $\mathcal{M}^\perp$.

In [28], the authors connect the indicator vectors $\boldsymbol{u}_i$s with the space $\mathcal{M}$. In particular, we have

**Proposition 2.1** ([28]). *All eigenvalues of $\boldsymbol{G}$ lie in the interval $(-1, 1]$. Furthermore, the nonnegative vectors $\{\tilde{\boldsymbol{D}}^{\frac{1}{2}} \boldsymbol{u}_i / \|\tilde{\boldsymbol{D}}^{\frac{1}{2}} \boldsymbol{u}_i\|\}_{1 \leq i \leq m}$ form an orthonormal basis of $\mathcal{M}$.*

Notice that we have the decomposition $\boldsymbol{H} = \boldsymbol{H}_{\mathcal{M}} + \boldsymbol{H}_{\mathcal{M}^\perp}$ for any matrix $\boldsymbol{H} := [\boldsymbol{h}_1, \boldsymbol{h}_2, \ldots, \boldsymbol{h}_n] \in \mathbb{R}^{d \times n}$ with $\boldsymbol{H}_{\mathcal{M}} = \sum_{i=1}^{m} \boldsymbol{H} \boldsymbol{e}_i \boldsymbol{e}_i^\top$ and $\boldsymbol{H}_{\mathcal{M}^\perp} = \sum_{i=m+1}^{n} \boldsymbol{H} \boldsymbol{e}_i \boldsymbol{e}_i^\top$ s.t. $\langle \boldsymbol{H}_{\mathcal{M}}, \boldsymbol{H}_{\mathcal{M}^\perp} \rangle_F = \text{Trace}(\sum_{i=1}^{m} \boldsymbol{H} \boldsymbol{e}_i \boldsymbol{e}_i^\top (\sum_{j=m+1}^{n} \boldsymbol{H} \boldsymbol{e}_j \boldsymbol{e}_j^\top)^\top) = 0$, which implies $\|\boldsymbol{H}\|_F^2 = \|\boldsymbol{H}_{\mathcal{M}}\|_F^2 + \|\boldsymbol{H}_{\mathcal{M}^\perp}\|_F^2$.

### 2.1 EXISTING SMOOTHNESS NOTIONS OF NODE FEATURES

**Distance to the eigenspace $\mathcal{M}$.** The paper [28] studies the smoothness of node features $\boldsymbol{H} := [\boldsymbol{h}_1, \ldots, \boldsymbol{h}_n] \in \mathbb{R}^{d \times n}$ using their distance to the eigenspace $\mathcal{M}$ as a smoothness notion.

**Definition 2.2** ([28]). *Let $\mathbb{R}^d \otimes \mathcal{M}$ be the subspace of $\mathbb{R}^{d \times n}$ consisting of the sum $\sum_{i=1}^{m} \boldsymbol{w}_i \otimes \boldsymbol{e}_i$ where $\boldsymbol{w}_i \in \mathbb{R}^d$ and $\{\boldsymbol{e}_i\}_{i=1}^{m}$ is an orthonormal basis of the eigenspace $\mathcal{M}$. Then we define $\|\boldsymbol{H}\|_{\mathcal{M}^\perp}$ – the distance of node features $\boldsymbol{H}$ to the eigenspace $\mathcal{M}$ – as follows:*

$$\|\boldsymbol{H}\|_{\mathcal{M}^\perp} := \inf_{\boldsymbol{Y} \in \mathbb{R}^d \otimes \mathcal{M}} \|\boldsymbol{H} - \boldsymbol{Y}\|_F = \left\|\boldsymbol{H} - \sum_{i=1}^{m} \boldsymbol{H} \boldsymbol{e}_i \boldsymbol{e}_i^\top\right\|_F.$$

With the decomposition $\boldsymbol{H} = \boldsymbol{H}_{\mathcal{M}} + \boldsymbol{H}_{\mathcal{M}^\perp}$, $\|\cdot\|_{\mathcal{M}^\perp}$ can be related to $\|\cdot\|_F$ as follows:

$$\|\boldsymbol{H}\|_{\mathcal{M}^\perp} = \|\boldsymbol{H} - \boldsymbol{H}_{\mathcal{M}}\|_F = \|\boldsymbol{H}_{\mathcal{M}^\perp}\|_F. \tag{3}$$

**Dirichlet energy.** [5] studies the smoothness of node features using Dirichlet energy, defined as:

**Definition 2.3** ([5]). *Let $\tilde{\Delta} = \boldsymbol{I} - \boldsymbol{G}$ be the (augmented) normalized Laplacian, then the Dirichlet energy $\|\boldsymbol{H}\|_E$ of node features $\boldsymbol{H}$ is defined by $\|\boldsymbol{H}\|_E^2 := \text{Trace}(\boldsymbol{H} \tilde{\Delta} \boldsymbol{H}^\top)$.*

**Normalized Dirichlet energy.** [5] points out that the smoothness of node features $\boldsymbol{H}$ should be measured by the normalized Dirichlet energy $\text{Trace}(\boldsymbol{H} \tilde{\Delta} \boldsymbol{H}^\top)/\|\boldsymbol{H}\|_F^2$. This normalized measurement is essential when data comes from various sources with diverse measurement units or scales. By normalizing the measurement, we can mitigate biases resulting from these different scales.

## 2.2 Two existing theories of over-smoothing

Let $\lambda = \max\{|\lambda_i| \mid \lambda_i < 1\}$ be the second largest magnitude of $G$'s eigenvalues, and $s_l$ be the largest singular value of $W^l$ – the weight matrix of the $l^{th}$ GCL. The authors of [28] show that $\|H^l\|_{\mathcal{M}^\perp} \leq s_l\lambda\|H^{l-1}\|_{\mathcal{M}^\perp}$ under equation (2) when $\sigma$ is ReLU. Therefore, $\|H^l\|_{\mathcal{M}^\perp} \to 0$ as $l \to \infty$ provided $s_l\lambda < 1$ for each $l$, indicating the node feature converges to the eigenspace $\mathcal{M}$ and results in over-smoothing. A crucial step in the analysis in [28] is that $\|\sigma(Z)\|_{\mathcal{M}^\perp} \leq \|Z\|_{\mathcal{M}^\perp}$ for any matrix $Z$ when $\sigma$ is ReLU, i.e. ReLU reduces the distance to eigenspace $\mathcal{M}$. The authors of [28] have mentioned that it is hard to extend the above result to even leaky ReLU.

Instead of considering $\|H\|_{\mathcal{M}^\perp}$, the paper [5] shows that $\|H^l\|_E \leq s_l\lambda\|H^{l-1}\|_E$ under equation (2) when $\sigma$ is ReLU or leaky ReLU. Hence, $\|H^l\|_E \to 0$ as $l \to \infty$, implying over-smoothing of GCNs. Note that $\|H\|_{\mathcal{M}^\perp} = 0$ or $\|H^l\|_E = 0$ means the feature vectors are homogeneous across all graph nodes. In particular, the proof in [5] applies to GCN with both ReLU and leaky ReLU activation functions by establishing the inequality that $\|\sigma(Z)\|_E \leq \|Z\|_E$ for any matrix $Z$.

# 3 Effects of Activation Functions: A Geometric Characterization

In this section, we present a geometric relationship between the input and output vectors of ReLU or leaky ReLU. We use $\|H\|_{\mathcal{M}^\perp}$ as the smoothness notion for all subsequent analyses since we observe that $\|H\|_{\mathcal{M}^\perp}$ and $\|H\|_E$ are equivalent as seminorms:

**Proposition 3.1.** $\|H\|_{\mathcal{M}^\perp}$ and $\|H\|_E$ are two equivalent seminorms, i.e. there exist two constants $\alpha, \beta > 0$ s.t. $\alpha\|H\|_{\mathcal{M}^\perp} \leq \|H\|_E \leq \beta\|H\|_{\mathcal{M}^\perp}$ for any $H \in \mathbb{R}^{d \times n}$.

**ReLU.** Let $\sigma(x) = \max\{x, 0\}$ be the ReLU activation function. The first main result of this paper is that there is a high-dimensional sphere underlying the input and output of ReLU:

**Proposition 3.2** (ReLU). *For any $Z = Z_{\mathcal{M}} + Z_{\mathcal{M}^\perp} \in \mathbb{R}^{d \times n}$, let $H = \sigma(Z) = H_{\mathcal{M}} + H_{\mathcal{M}^\perp}$. Then $H_{\mathcal{M}^\perp}$ lies on the high-dimensional sphere centered at $Z_{\mathcal{M}^\perp}/2$ with radius*

$$r := \left(\|Z_{\mathcal{M}^\perp}/2\|_F^2 - \langle H_{\mathcal{M}}, H_{\mathcal{M}} - Z_{\mathcal{M}}\rangle_F\right)^{1/2}.$$

*In particular, $H_{\mathcal{M}^\perp}$ lies inside the ball centered at $Z_{\mathcal{M}^\perp}/2$ with radius $\|Z_{\mathcal{M}^\perp}/2\|_F$ and hence we have $\|H\|_{\mathcal{M}^\perp} \leq \|Z\|_{\mathcal{M}^\perp}$.*

**Leaky ReLU.** Now we consider leaky ReLU $\sigma_a(x) = \max\{x, ax\}$, where $0 < a < 1$ is a positive scalar. Similar to ReLU, we have the following result for leaky ReLU:

**Proposition 3.3** (Leaky ReLU). *For any $Z = Z_{\mathcal{M}} + Z_{\mathcal{M}^\perp} \in \mathbb{R}^{d \times n}$, let $H = \sigma_a(Z) = H_{\mathcal{M}} + H_{\mathcal{M}^\perp}$. Then $H_{\mathcal{M}^\perp}$ lies on the high-dimensional sphere centered at $(1 + a)Z_{\mathcal{M}^\perp}/2$ with radius*

$$r_a := \left(\|(1-a)Z_{\mathcal{M}^\perp}/2\|_F^2 - \langle H_{\mathcal{M}} - Z_{\mathcal{M}}, H_{\mathcal{M}} - aZ_{\mathcal{M}}\rangle_F\right)^{1/2}.$$

*In particular, $H_{\mathcal{M}^\perp}$ lies inside the ball centered at $(1 + a)Z_{\mathcal{M}^\perp}/2$ with radius $\|(1-a)Z_{\mathcal{M}^\perp}/2\|_F$ and hence we see that $a\|Z\|_{\mathcal{M}^\perp} \leq \|H\|_{\mathcal{M}^\perp} \leq \|Z\|_{\mathcal{M}^\perp}$.*

## 3.1 Implications of the above geometric characterizations

Propositions 3.2 and 3.3 imply that the precise location of $H_{\mathcal{M}^\perp}$ (or the smoothness $\|H_{\mathcal{M}^\perp}\|_F = \|H\|_{\mathcal{M}^\perp}$) depends on the center and the radius, $r$ or $r_a$, of the spheres described in the respective propositions. Given a fixed $Z_{\mathcal{M}^\perp}$, the center of the spheres remains unchanged, and their radii $r$ and $r_a$ are only affected by changes in $Z_{\mathcal{M}}$. This observation motivates us to investigate ***how changes in $Z_{\mathcal{M}}$ impact $\|H\|_{\mathcal{M}^\perp}$, i.e. the smoothness of node features***.

However, from Propositions 3.2 and 3.3, we see that both ReLU and leaky ReLU reduce the distance of node features to the eigenspace $\mathcal{M}$, i.e. $\|H\|_{\mathcal{M}^\perp} \leq \|Z\|_{\mathcal{M}^\perp}$. Moreover, the above inequality is independent of $Z_{\mathcal{M}}$; consider two node features $Z, Z' \in \mathbb{R}^{d \times n}$ s.t. $Z_{\mathcal{M}^\perp} = Z'_{\mathcal{M}^\perp}$ but $Z_{\mathcal{M}} \neq Z'_{\mathcal{M}}$. Let $H, H'$ be the output of $Z, Z'$ via ReLU or leaky ReLU, respectively. Then we have the inequalities $\|H\|_{\mathcal{M}^\perp} \leq \|Z\|_{\mathcal{M}^\perp}$ and $\|H'\|_{\mathcal{M}^\perp} \leq \|Z'\|_{\mathcal{M}^\perp}$. Since $Z_{\mathcal{M}^\perp} = Z'_{\mathcal{M}^\perp}$ implies that $\|Z\|_{\mathcal{M}^\perp} = \|Z'\|_{\mathcal{M}^\perp}$, we deduce that $\|H'\|_{\mathcal{M}^\perp} \leq \|Z\|_{\mathcal{M}^\perp}$. In other words, when $Z_{\mathcal{M}^\perp} = Z'_{\mathcal{M}^\perp}$ is fixed, ***changing $Z_{\mathcal{M}}$ to $Z'_{\mathcal{M}}$ can change the smoothness of the output features but can not change the fact that ReLU and leaky ReLU smooth node features***; we demonstrate this result in Fig. 1a) in Section 4.1. In contrast, ***if one considers the normalized smoothness, we find that adjusting $Z_{\mathcal{M}}$ can result in a less smooth output***; we will discuss this in Section 4.1.

## 4 HOW ADJUSTING $\boldsymbol{Z}_{\mathcal{M}}$ AFFECTS THE SMOOTHNESS OF THE OUTPUT

Throughout this section, we let $\boldsymbol{Z}$ and $\boldsymbol{H}$ be the input and output of ReLU or leaky ReLU. The smoothness notions based on the feature vectors' distance to the eigenspace $\mathcal{M}$ or their Dirichlet energy do not account for the magnitude of each dimension of the learned node features; the authors of [5] point out that analyzing the normalized smoothness of feature vectors $\boldsymbol{Z}$, $\|\boldsymbol{Z}\|_E/\|\boldsymbol{Z}\|_F$, remains an open problem. Furthermore, these two smoothness notions aggregate the smoothness of node features across all feature dimensions; when the magnitude of some dimensions is much larger than others, the smoothness will be dominated by these dimensions.

Motivated by the discussion in Section 3.1, we study *the disparate effects of adjusting $\boldsymbol{Z}_{\mathcal{M}}$ on the normalized and unnormalized smoothness* in this section. For the sake of simplicity, we assume the graph is connected ($m = 1$); all the following results can be extended to the graph with multiple connected components easily. Due to the equivalence between seminorms $\|\cdot\|_{\mathcal{M}}$ and $\|\cdot\|_E$, we introduce the following definition of the dimension-wise normalized smoothness of node features.

**Definition 4.1.** *Let $\boldsymbol{Z} \in \mathbb{R}^{d \times n}$ be the features over $n$ nodes with $\boldsymbol{z}^{(i)} \in \mathbb{R}^n$ ($i = 1, \ldots, d$) being the $i^{th}$ row vector of $\boldsymbol{Z}$, i.e. the $i^{th}$ dimension of the features over all nodes. Then we define the normalized smoothness of $\boldsymbol{z}^{(i)}$ as $s(\boldsymbol{z}^{(i)}) \coloneqq \|\boldsymbol{z}_{\mathcal{M}}^{(i)}\|/\|\boldsymbol{z}^{(i)}\|$ with $\boldsymbol{z}_{\mathcal{M}}^{(i)}$ being the projection of $\boldsymbol{z}^{(i)}$ onto the eigenspace $\mathcal{M}$, where we set $s(\boldsymbol{z}^{(i)}) = 1$ when $\boldsymbol{z}^{(i)} = \boldsymbol{0}$.*

The graph is connected ($m = 1$) implies that $\boldsymbol{z}_{\mathcal{M}}^{(i)} = \langle \boldsymbol{z}^{(i)}, \boldsymbol{e}_1 \rangle \boldsymbol{e}_1$ and $\|\boldsymbol{z}_{\mathcal{M}}^{(i)}\| = |\langle \boldsymbol{z}^{(i)}, \boldsymbol{e}_1 \rangle|$, which will be used in the following analysis. Without ambiguity, we drop the index and simply write $\boldsymbol{z}$ for $\boldsymbol{z}^{(i)}$ and $\boldsymbol{e}$ for $\boldsymbol{e}_1$ – the only eigenvector of $\boldsymbol{G}$ associated with the eigenvalue 1. Moreover, we have

$$s(\boldsymbol{z}) = \frac{\|\boldsymbol{z}_{\mathcal{M}}\|}{\|\boldsymbol{z}\|} = \frac{|\langle \boldsymbol{z}, \boldsymbol{e} \rangle|}{\|\boldsymbol{z}\|} = \frac{|\langle \boldsymbol{z}, \boldsymbol{e} \rangle|}{\|\boldsymbol{z}\| \cdot \|\boldsymbol{e}\|} \Rightarrow 0 \leq s(\boldsymbol{z}) \leq 1, \tag{4}$$

It is evident that *the larger the value $s(\boldsymbol{z})$ is, the smoother the node feature $\boldsymbol{z}$ is*[1]. In fact, we have

$$s(\boldsymbol{z})^2 + \left(\frac{\|\boldsymbol{z}\|_{\mathcal{M}^\perp}}{\|\boldsymbol{z}\|}\right)^2 = \frac{\|\boldsymbol{z}_{\mathcal{M}}\|^2}{\|\boldsymbol{z}\|^2} + \frac{\|\boldsymbol{z}_{\mathcal{M}^\perp}\|^2}{\|\boldsymbol{z}\|^2} = 1,$$

where we see $\|\boldsymbol{z}\|_{\mathcal{M}^\perp}/\|\boldsymbol{z}\|$ decreases as $s(\boldsymbol{z})$ increases.

To discuss how the smoothness $s(\boldsymbol{h}) = s(\sigma(\boldsymbol{z}))$ or $s(\sigma_a(\boldsymbol{z}))$ can be adjusted by changing $\boldsymbol{z}_{\mathcal{M}}$, we consider the function $\boldsymbol{z}(\alpha) = \boldsymbol{z} - \alpha\boldsymbol{e}$ where $\boldsymbol{z}(0) = \boldsymbol{z}$. It is clear that

$$\boldsymbol{z}(\alpha)_{\mathcal{M}^\perp} = \boldsymbol{z}_{\mathcal{M}^\perp} \text{ and } \boldsymbol{z}(\alpha)_{\mathcal{M}} = \boldsymbol{z}_{\mathcal{M}} - \alpha\boldsymbol{e},$$

where we see that $\alpha$ only alters $\boldsymbol{z}_{\mathcal{M}}$ while preserves $\boldsymbol{z}_{\mathcal{M}^\perp}$. Moreover, it is evident that

$$s(\boldsymbol{z}(\alpha)) = \sqrt{1 - \frac{\|\boldsymbol{z}(\alpha)_{\mathcal{M}^\perp}\|^2}{\|\boldsymbol{z}(\alpha)\|^2}} = \sqrt{1 - \frac{\|\boldsymbol{z}_{\mathcal{M}^\perp}\|^2}{\|\boldsymbol{z}(\alpha)\|^2}}.$$

It follows that $s(\boldsymbol{z}(\alpha)) = 1$ if and only if $\boldsymbol{z}_{\mathcal{M}^\perp} = \boldsymbol{0}$ (include the case $\boldsymbol{z} = \boldsymbol{0}$), showing that when $\boldsymbol{z}_{\mathcal{M}^\perp} = \boldsymbol{0}$, the vector $\boldsymbol{z}$ is the smoothest one.

### 4.1 THE DISPARATE EFFECTS OF $\alpha$ ON $\|\cdot\|_{\mathcal{M}^\perp}$ AND $s(\cdot)$: EMPIRICAL RESULTS

Let us conduct a simple empirical study to investigate possible values that the (unnormalized) smoothness $\|\sigma(\boldsymbol{z}(\alpha))\|_{\mathcal{M}^\perp}$, $\|\sigma_a(\boldsymbol{z}(\alpha))\|_{\mathcal{M}^\perp}$ and the normalized smoothness $s(\sigma(\boldsymbol{z}(\alpha)))$, $s(\sigma_a(\boldsymbol{z}(\alpha)))$ can take when $\alpha$ varies. We denote $\boldsymbol{z}_\alpha \coloneqq \boldsymbol{z}(\alpha) = \boldsymbol{z} - \alpha\boldsymbol{e}$. We consider a connected synthetic graph with 100 nodes, and each node is assigned a random degree between 2 to 10. Then we assign an initial node feature $\boldsymbol{z} \in \mathbb{R}^{100}$, sampled uniformly on the interval $[-1.5, 1.5]$, to the graph with each node feature being a scalar. Also, we compute $\boldsymbol{e}$ by the formula $\boldsymbol{e} = \tilde{\boldsymbol{D}}^{\frac{1}{2}}\boldsymbol{u}/\|\tilde{\boldsymbol{D}}^{\frac{1}{2}}\boldsymbol{u}\|$ from Proposition 2.1 where $\boldsymbol{u} \in \mathbb{R}^{100}$ is the vector whose entries are all ones

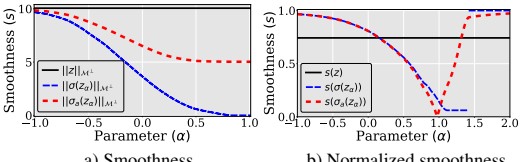

a) Smoothness     b) Normalized smoothness

Figure 1: Contrasting the effects of varying parameter $\alpha$ on the smoothness and normalized smoothness of output features $\sigma(\boldsymbol{z}_\alpha)$ and $\sigma_a(\boldsymbol{z}_\alpha)$. The discontinuity of $s(\sigma(\boldsymbol{z}_\alpha))$ in b) comes from the definition of normalized smoothness. Note that $s(\boldsymbol{z}) = 1$ if $\boldsymbol{z} = \boldsymbol{0}$, and $\sigma(\boldsymbol{z}_\alpha)$ can become $\boldsymbol{0}$ when $\alpha$ is large enough.

---

[1]Here, $\boldsymbol{z} \in \mathbb{R}^n$ is a vector whose $i^{th}$ entry is the 1D feature associated with node $i$.

and $\tilde{D}$ is the (augmented) degree matrix of the graph. We examine two different smoothness notions for the input node features $z$ and the output node features $\sigma(z_\alpha)$ and $\sigma_a(z_\alpha)$, where the smoothness is measured for various values of the smoothness control parameter $\alpha \in [-1.5, 1.5]$. In Fig. 1a), we study the smoothness measured by $\|\cdot\|_{\mathcal{M}^\perp}$; we see that $\|\sigma(z_\alpha)\|_{\mathcal{M}^\perp}$ and $\|\sigma_a(z_\alpha)\|_{\mathcal{M}^\perp}$ are always no greater than $\|z\|_{\mathcal{M}^\perp}$. This coincides with the discussion in Section 3.1; adjusting the projection of $z$ onto the eigenspace $\mathcal{M}$ can not change the fact that $\|\sigma(z_\alpha)\|_{\mathcal{M}^\perp} \leq \|z\|_{\mathcal{M}^\perp}$ and $\|\sigma_a(z_\alpha)\|_{\mathcal{M}^\perp} \leq \|z\|_{\mathcal{M}^\perp}$. Nevertheless, an interesting result is that ***altering the eigenspace projection can adjust the smoothness of the output***: notice that altering the eigenspace projection does not change its distance to $\mathcal{M}$, i.e. the smoothness of the input is unchanged, but the smoothness of the output after activation function can be changed.

In contrast, when we study the normalized smoothness measured by $s(\cdot)$ in Fig. 1b), we find that $s(\sigma(z(\alpha)))$ and $s(\sigma_a(z(\alpha)))$ can be adjusted by $\alpha$ to the values smaller than $s(z)$. More precisely, we see that by adjusting $\alpha$, $s(\sigma(z(\alpha)))$ and $s(\sigma_a(z(\alpha)))$ can achieve most of the values in $[0, 1]$. In other words, both smoother and less smooth features can be obtained.

## 4.2 THEORETICAL RESULTS ON THE SMOOTH EFFECTS OF RELU AND LEAKY RELU

In this subsection, we establish theoretical understandings of the above empirical findings on the achievable smoothness of $\sigma(z(\alpha))$ and $\sigma_a(z(\alpha))$ by adjusting $\alpha$ – shown in Fig. 1. Notice that if $z_{\mathcal{M}^\perp} = 0$, the inequalities presented in Proposition 3.2 and Proposition 3.3 indicate that $\|\sigma(z(\alpha))\|_{\mathcal{M}^\perp}$ and $\|\sigma_a(z(\alpha))\|_{\mathcal{M}^\perp}$ vanish. So we have $s(\sigma(z(\alpha))) = 1$ for any $\alpha$ when $z_{\mathcal{M}^\perp} = 0$. Then we may assume $z_{\mathcal{M}^\perp} \neq 0$ for the following study.

**Proposition 4.2** (ReLU). *Suppose $z_{\mathcal{M}^\perp} \neq 0$. Let $h(\alpha) = \sigma(z(\alpha))$ with $\sigma$ being ReLU, then*

$$\min_\alpha s(h(\alpha)) = \sqrt{\frac{\sum_{x_i = \max x} d_i}{\sum_{j=1}^n d_j}} \quad and \quad \max_\alpha s(h(\alpha)) = 1,$$

*where $x := \tilde{D}^{-\frac{1}{2}} z$, $\max x = \max_{1 \leq i \leq n} x_i$, and $\tilde{D}$ is the augmented degree matrix with diagonals $d_1, d_2, \ldots, d_n$. In particular, the normalized smoothness $s(h(\alpha))$ is monotone increasing as $\alpha$ decreases whenever $\alpha < \|\tilde{D}^{\frac{1}{2}} u_n\| \max x$ and it has range $[\min_\alpha s(h(\alpha)), 1]$.*

**Proposition 4.3** (Leaky ReLU). *Suppose $z_{\mathcal{M}^\perp} \neq 0$. Let $h(\alpha) = \sigma_a(z(\alpha))$ with $\sigma_a$ being leaky ReLU, then 1) $\min_\alpha s(h(\alpha)) = 0$, and 2) $\sup_\alpha s(h(\alpha)) = 1$. In particular, $s(h(\alpha))$ has range $[0, 1)$.*

**Remark 4.4.** *Proposition 4.3 also holds for other variants of ReLU, like ELU[2] and SELU[3]. Please see Appendix C for details.*

As a brief summary of Propositions 3.2, 3.3, 4.2, and 4.3, we provide the following Corollary, which qualitatively explains the empirical results in Figure 1.

**Corollary 4.5.** *Suppose $z_{\mathcal{M}^\perp} \neq 0$. Let $h(\alpha) = \sigma(z(\alpha))$ or $\sigma_a(z(\alpha))$ with $\sigma$ being ReLU and $\sigma_a$ being leaky ReLU. Then we have $\|z\|_{\mathcal{M}^\perp} \geq \|h(\alpha)\|_{\mathcal{M}^\perp}$ for any $\alpha \in \mathbb{R}$; however, $s(h(\alpha))$ is not always smaller than $s(z)$. In particular, $s(h(\alpha))$ can be smaller than, larger than, or equal to $s(z)$ for different values of $\alpha$.*

Propositions 4.2 and 4.3, as well as Corollary 4.5, provide a theoretical basis for the empirical results presented in Fig. 1. Moreover, our results indicate that for any given vector $z$, altering its projection $z_{\mathcal{M}}$ can effectively change both the unnormalized and the normalized smoothness of the output vector $h = \sigma(z)$ or $\sigma_a(z)$. In particular, the normalized smoothness of the output vector $h = \sigma(z)$ or $\sigma_a(z)$ can be adjusted to any value in the range shown in Proposition 4.2 and 4.3. This provides us with insights to design algorithms to control the smoothness of feature vectors to improve the performance of GCN and we will discuss this in the next section.

## 5 CONTROLLING SMOOTHNESS OF NODE FEATURES

For a given graph node classification task, we usually do not know how smooth features are ideal, which has been empirically confirmed in [28] by studying a smoothness-related quantity. Nevertheless, our established theory indicates that both the normalized and unnormalized smoothness of the output of each GCL can be adjusted by altering the input's projection onto the eigenspace $\mathcal{M}$. Motivated by our theory, we propose the following learnable smoothness control term to modulate the smoothness of each dimension of the learned node feature vectors:

---

[2]The ELU function is defined by $f(x) = \max(x, 0) + \min(0, a \cdot (e^x - 1))$ where $a > 0$.
[3]The SELU function is defined by $f(x) = c(\max(x, 0) + \min(0, a \cdot (e^x - 1)))$ where $a, c > 0$.

$$\boldsymbol{B}_{\boldsymbol{\alpha}}^l = \sum_{i=1}^m \boldsymbol{\alpha}_i^l \boldsymbol{e}_i^\top, \tag{5}$$

where $l$ is the layer index, $\{\boldsymbol{e}_i\}_{i=1}^m$ is the orthonormal basis of the eigenspace $\mathcal{M}$, and $\boldsymbol{\alpha}^l := \{\boldsymbol{\alpha}_i^l\}_{i=1}^m$ is a collection of learnable vectors with $\boldsymbol{\alpha}_i^l \in \mathbb{R}^d$ being approximated by a multi-layer perceptron (MLP). The detailed configuration of $\boldsymbol{\alpha}_i^l$ will be specified in each experiment later. One can see that $\boldsymbol{B}_{\boldsymbol{\alpha}}^l$ always lies in $\mathbb{R}^d \otimes \mathcal{M}$. We integrate SCT into GCL, resulting in

$$\boldsymbol{H}^l = \sigma(\boldsymbol{W}^l \boldsymbol{H}^{l-1} \boldsymbol{G} + \boldsymbol{B}_{\boldsymbol{\alpha}}^l). \tag{6}$$

We call the corresponding model GCN with a smoothness control term (GCN-SCT). Again, the idea is that *we change the component in eigenspace to control the smoothness of node features*. In particular, each dimension of the output $\boldsymbol{H}^l$ can be smoother, less smooth, or the same as that of $\boldsymbol{H}^{l-1}$ in terms of normalized smoothness, though $\boldsymbol{H}^l$ gets closer to the eigenspace $\mathcal{M}$ than $\boldsymbol{H}^{l-1}$.

Next, we provide some details about the proposed SCT. To design SCT, we introduce a learnable matrix $\boldsymbol{A}^l \in \mathbb{R}^{d \times m}$ for layer $l$, whose columns are $\boldsymbol{\alpha}_i^l$, where $m$ is the dimension of the eigenspace $\mathcal{M}$ and $d$ is the dimension of the features. We observed in our experiments that the SCT performs best when informed by degree pooling over the subcomponents of the graph. The matrix of the orthogonal basis vectors, denoted by $\boldsymbol{Q} := [\boldsymbol{e}_1, \ldots, \boldsymbol{e}_m] \in \mathbb{R}^{n \times m}$, is used to perform pooling $\boldsymbol{H}^l \boldsymbol{Q}$ for input $\boldsymbol{H}^l$. In particular, we let $\boldsymbol{A}^l = \boldsymbol{W} \odot (\boldsymbol{H}^l \boldsymbol{Q})$, where $\boldsymbol{W} \in \mathbb{R}^{d \times m}$ is learnable and performs pooling over $\boldsymbol{H}^l$ using the eigenvectors $\boldsymbol{Q}$. The second architecture uses a residual connection with hyperparameter $\beta_l = \log(\theta/l + 1)$, following GCNII [7], and learnable matrices $\boldsymbol{W}_0, \boldsymbol{W}_1 \in \mathbb{R}^{d \times d}$ and the softmax function $\phi$. Resulting in $\boldsymbol{A}^l = \phi(\boldsymbol{H}^l \boldsymbol{Q}) \odot (\beta_l \boldsymbol{W}_0 \boldsymbol{H}^0 \boldsymbol{Q} + (1 - \beta_l) \boldsymbol{W}_1 \boldsymbol{H}^l \boldsymbol{Q})$. In Section 6, we use the first architecture for GCN-SCT because GCN uses only $\boldsymbol{H}^l$ information at each layer. We use the second architecture for GCNII-SCT and EGNN-SCT which use both $\boldsymbol{H}^0$ and $\boldsymbol{H}^l$ information at each layer.

## 5.1 Integrating SCT into other GCN-style models

In this subsection, we present other usages of the proposed SCT. Due to the page limit, we carefully select two other most representative usages of the proposed SCT. The first example is GCNII [7], GCNII extends GCN to express an arbitrary polynomial filter rather than the Laplacian polynomial filter and has been shown to achieve state-of-the-art performance among GCN-style models on various benchmark tasks [7; 25], and we aim to show that the proposed SCT can even benefits node classification for the GCN-style model that achieves state-of-the-art performance on many node classification tasks. The second example is energetic GNN (EGNN) [42], which controls the smoothness of node features by constraining the lower and upper bounds of the Dirichlet energy of node features and assuming the activation function is linear. In this case, we aim to show that our established theoretical understanding of the role of activation functions and the proposed SCT can boost the performance of EGNN with consideration of nonlinear activation functions.

**GCNII:** Each GCNII layer uses a residual connection to the initial layer $\boldsymbol{H}^0$ and given as follows:

$$\boldsymbol{H}^l = \sigma\big(((1 - \alpha_l)\boldsymbol{H}^{l-1}\boldsymbol{G} + \alpha_l \boldsymbol{H}^0)((1 - \beta_l)\boldsymbol{I} + \beta_l \boldsymbol{W}^l)\big), \ \text{ where } \alpha_l, \beta_l \in (0, 1) \text{ are learnable.}$$

We integrate SCT $\boldsymbol{B}_{\boldsymbol{\alpha}}^l$ into GCNII, resulting in the following GCNII-SCT layers:

$$\boldsymbol{H}^l = \sigma\big(((1 - \alpha_l)\boldsymbol{H}^{l-1}\boldsymbol{G} + \alpha_l \boldsymbol{H}^0)((1 - \beta_l)\boldsymbol{I} + \beta_l \boldsymbol{W}^l) + \boldsymbol{B}_{\boldsymbol{\alpha}}^l\big),$$

where the residual connection and identity mapping are consistent with GCNII.

**EGNN:** Each EGNN layer can be written as follows:

$$\boldsymbol{H}^l = \sigma\big(\boldsymbol{W}^l(c_1 \boldsymbol{H}^0 + c_2 \boldsymbol{H}^{l-1} + (1 - c_{\min})\boldsymbol{H}^{l-1}\boldsymbol{G})\big), \tag{7}$$

where $c_1, c_2$ are learnable weights that satisfy $c_1 + c_2 = c_{\min}$ with $c_{\min}$ being a hyperparameter. To constrain Dirichlet energy, EGNN initializes trainable weights $\boldsymbol{W}^l$ as a diagonal matrix with explicit singular values and regularizes them to keep the orthogonality during the model training. Ignoring the activation function $\sigma$, $\boldsymbol{H}^l$ – node features at layer $l$ of EGNN satisfies:

$$c_{\min}\|\boldsymbol{H}^0\|_E \leq \|\boldsymbol{H}^l\|_E \leq c_{\max}\|\boldsymbol{H}^0\|_E,$$

where $c_{\max}$ is the square of the maximal singular value of the initialization of $\boldsymbol{W}^1$.

Similarly, we modify EGNN to result in the following EGNN-SCT layer:

$$\boldsymbol{H}^l = \sigma\big(\boldsymbol{W}^l((1 - c_{\min})\boldsymbol{H}^{l-1}\boldsymbol{G} + c_1 \boldsymbol{H}^0 + c_2 \boldsymbol{H}^{l-1}) + \boldsymbol{B}_{\boldsymbol{\alpha}}^l\big),$$

where everything remains the same as EGNN except that we add our proposed SCT $\boldsymbol{B}_{\boldsymbol{\alpha}}^l$.

# 6 EXPERIMENTS

In this section, we comprehensively demonstrate the effects of SCT – in the three most representative GCN-style models discussed in Section 5 – using various node classification benchmarks. The purpose of all experiments in this section is to verify the efficacy of the proposed SCT – motivated by our theoretical results – for GCN-style models. Exploring the effects of SCT on non-GCN-style models and pushing for state-of-the-art accuracy is an interesting future direction. We consider the citation datasets (Cora, Citeseer, PubMed, Coauthor-Physics, Ogbn-arxiv), web knowledge-base datasets (Cornell, Texas, Wisconsin), and Wikipedia network datasets (Chameleon, Squirrel). We provide additional dataset details in Appendix D.1. We implement baseline GCN and GCNII (without weight sharing) using PyG [12]. Baseline EGNN is implemented using the public code[4].

## 6.1 NODE FEATURE TRAJECTORY

We visualize the trajectory of the node features, following [28], for a graph with two nodes connected by an edge and 1D node feature. In this case, equation (6) becomes $h^1 = \sigma(wh^0 G + b_\alpha)$, where $w = 1.2$ in our experiment, $h^0, h^1, b_\alpha \in \mathbb{R}^2$, and $G \in \mathbb{R}^{2 \times 2}$. We use a matrix $G = [0.592, 0.194; 0.194, 0.908]$ whose largest eigenvalue is 1. Twenty initial node feature vectors $h^0$ are sam-

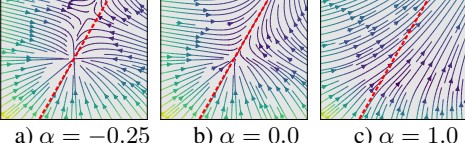

a) $\alpha = -0.25$   b) $\alpha = 0.0$   c) $\alpha = 1.0$

Figure 2: Node feature trajectories, with colorized magnitude, for varying smoothness control parameter $\alpha$. For classical GCN b), the node features converge to the eigenspace $\mathcal{M}$ (red dashed line).

pled evenly in the domain $[-1, 1] \times [-1, 1]$. Fig. 2 shows the trajectories in relation to the eigenspace $\mathcal{M}$ (red dashed line). In Fig 2a), one can see that some trajectories do not directly converge to $\mathcal{M}$. In Fig. 2b) when $\alpha = 0.0$, GCL is recovered and all trajectories converge to $\mathcal{M}$. In Fig. 2c), large values of $\alpha$ enable the features to significantly deviate from $\mathcal{M}$ initially. We observe that the parameter $\alpha$ can effectively change the trajectory of features.

| | Layers | 2 | 4 | 16 | 32 |
|---|---|---|---|---|---|
| Cora | GCN/GCN-SCT | 81.1/**82.9** | 80.4/**82.8** | 64.9/**71.4** | 60.3/**67.2** |
| | GCNII/GCNII-SCT | 82.2/**83.8** | 82.6/**84.3** | 84.6/**84.8** | 85.4/**85.5** |
| | EGNN/EGNN-SCT | 83.2/**84.1** | 84.2/**84.5** | **85.4**/83.3 | **85.3**/82.0 |
| Citeseer | GCN/GCN-SCT | **70.3**/69.9 | 67.6/**67.7** | 18.3/**55.4** | 25.0/**51.0** |
| | GCNII/GCNII-SCT | 68.2/**72.8** | 68.9/**72.8** | 72.9/**73.8** | **73.4**/73.4 |
| | EGNN/EGNN-SCT | 72.0/**73.1** | 71.9/**72.0** | 72.4/**72.6** | 72.3/**72.9** |
| PubMed | GCN/GCN-SCT | 79.0/**79.8** | 76.5/**78.4** | 40.9/**76.1** | 22.4/**77.0** |
| | GCNII/GCNII-SCT | 78.2/**79.7** | 78.8/**80.1** | 80.2/**80.7** | 79.8/**80.7** |
| | EGNN/EGNN-SCT | 79.2/**79.8** | 79.5/**80.4** | 80.1/**80.3** | 80.0/**80.4** |
| Coauthor-Physics | GCN/GCN-SCT | 92.4/**92.6** $\pm$ **1.6** | 92.1/**92.5** $\pm$ **5.9** | 13.5/**50.9** $\pm$ **15.0** | 13.1/**43.6** $\pm$ **16.0** |
| | GCNII/GCNII-SCT | 92.5/**94.4** $\pm$ **0.4** | 92.9/**94.2** $\pm$ **0.3** | 92.9/**93.7** $\pm$ **0.7** | 92.9/**94.1** $\pm$ **0.3** |
| | EGNN/EGNN-SCT | 92.6/**93.9** $\pm$ **0.7** | 92.9/**94.1** $\pm$ **0.4** | 93.1/**94.0** $\pm$ **0.7** | 93.3/**93.8** $\pm$ **1.3** |
| Ogbn-arxiv | GCN/GCN-SCT | 70.4/**72.1** $\pm$ **0.3** | 71.7/**72.7** $\pm$ **0.3** | 70.6/**72.3** $\pm$ **0.2** | 68.5/**72.3** $\pm$ **0.3** |
| | GCNII/GCNII-SCT | 70.1/**72.0** $\pm$ **0.3** | 71.4/**72.2** $\pm$ **0.2** | 71.5/**72.4** $\pm$ **0.3** | 70.5/**72.1** $\pm$ **0.3** |
| | EGNN/EGNN-SCT | 68.4/**68.5** $\pm$ **0.6** | 71.1/**71.3** $\pm$ **0.5** | 72.7/**72.8** $\pm$ **0.5** | **72.7**/72.3 $\pm$ 0.5 |

Table 1: Accuracy for models of varying depth. We notice that vanishing gradients occur but not over-smoothing for the accuracy drop using GCN-SCT with 16 or 32 layers. For Cora, Citeseer, and PubMed, we use a fixed split with a single forward pass following [7]; only test accuracy is available in these experiments. For Coauthor-Physics and Ogbn-arxiv, we use the splits from [42]; both test accuracy and standard deviation are reported. The baseline results are copied from [7; 42], in which the standard deviation was not reported. (Unit:%)

## 6.2 BASELINE COMPARISONS FOR NODE CLASSIFICATION

**Citation networks.** We compare the three representative models discussed in Section 5, of different depths, with and without SCT in Table 1. This task uses the citation datasets with fixed splits from [37] for Cora, Citeseer, and Pubmed and 10 fixed cross-validation splits from [42] for Coauthor-Physics and Ogbn-arxiv; a detailed description of these datasets and splits are provided in Appendix D. We report the cross-validation mean and standard deviation for Coauthor-Physics and Ogbn-arxiv. Following [7], we minimize the negative log-likelihood loss using the Adam optimizer [21], with 1500 maximum epochs, and 100 epochs of patience. A grid search for possible hyperparameters is listed in Table 6 in Appendix D. We accelerate the hyperparameter search by applying a Bayesian meta-learning algorithm [3] which minimizes the validation loss, and we run the search for 200 iterations per model. In particular, Table 1 presents the best test accuracy between ReLU and leaky ReLU for GCN, GCNII, and all three models with SCT[5]. For the baseline EGNN, we follow [42] using SReLU, a particular activation used for EGNN in [42]. These results show that SCT can boost the classification accuracy of baseline models; in particular, the improvement can be remarkable for GCN and GCNII. However, EGNN-SCT (using ReLU or leaky ReLU) performs occasionally worse

---

[4]https://github.com/Kaixiong-Zhou/EGNN
[5]A comparison of the results using ReLU and leaky ReLU is presented in Appendix D.

than EGNN (using SReLU), and this is because of the choice of activation functions. In Appendix D.3, we report the results of EGNN-SCT using SReLU, showing that EGNN-SCT outperforms EGNN in all tasks. In fact, SReLU is a shifted version of ReLU, and our theory for ReLU applies to SReLU as well. The model size and computational time are reported in Table 5 in the appendix.

Table 1 also shows that even with SCT, the accuracy of GCN drops when the depth is 16 or 32. This motivates us to investigate the smoothness of the node features learned by GCN and GCN-SCT. Fig. 3 plots the heatmap of the normalized smoothness of each dimension of the learned node features learned by GCN and GCN-SCT with 32 layers for Citeseer node classification. In these plots, the horizontal and vertical dimensions denote the feature dimension and the layer of the model, respectively. We notice that the normalized smoothness of each dimension of the features – from layers 14 to 32 learned by GCN – closes to 1, confirming that deep GCN learns homogeneous features. In contrast, the features learned by GCN-SCT are inhomogeneous, as shown in Fig. 3b). Therefore, we believe the performance degradation of deep GCN-SCT is due to other factors. In particular, compared to GCNII/GCNII-SCT and EGNN/EGNN-SCT, GCN-SCT does not use skip connections, which is known to help avoid vanishing gradients in training deep neural networks [18; 19]. In Appendix D.3, we show that training GCN and GCN-SCT does suffer from the vanishing gradient issue; however, the other models do not. Besides Citeseer, we notice similar behavior occurs for training GCN and GCN-SCT for Cora and Coauthor-Physics node classification tasks.

**Other datasets.** We further compare different models trained on different datasets using 10-fold cross-validation and fixed $48/32/20\%$ splits following [29]. Table 2 compares GCN and GCNII with and without SCT, using leaky ReLU, for classifying five heterophilic node classification datasets. We exclude EGNN as these heterophilic datasets are not considered in [42]. We report the average test accuracy of GCN and GCNII from [7]. We tune all other models using a Bayesian meta-learning algorithm to maximize the mean validation accuracy. We report the best test accuracy for each model of depth searched over the set $\{2, 4, 8, 16, 32\}$. SCT can significantly improve the classification accuracy of the baseline models. Table 2 also contrasts the computational time (on Tesla T4 GPUs from Google Colab) per epoch of models that achieve the best test accuracy; the models using SCT can even save computational time to achieve the best accuracy which is because the best accuracy is achieved at a moderate depth (Table 11 in Appendix D.4 lists the mean and standard deviation for the test accuracies on all five datasets. Table 12 in Appendix D.4 lists the computational time per epoch for each model of depth 8, showing that using SCT only takes a small amount of computational overhead.

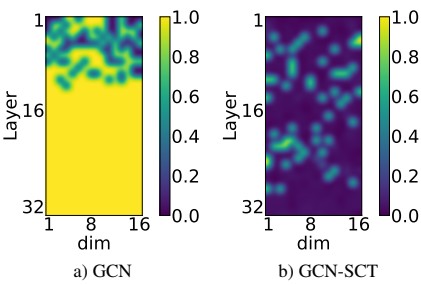

a) GCN      b) GCN-SCT

Figure 3: The normalized smoothness – of each dimension of the feature vectors at a given layer – for a) GCN and b) GCN-SCT on the Citeseer dataset with 32 layers and 16 hidden dimensions. GCN features become entirely smooth since layer 14, while GCN-SCT controls the smoothness for each feature at any depth. Horizontal and vertical axes represent the index of the feature dimension and the intermediate layer, respectively.

| | Cornell | Texas | Wisconsin | Chameleon | Squirrel |
|---|---|---|---|---|---|
| | 52.70/**55.95** (0.7/1.8) | 52.16/**62.16** (0.7/0.8) | 45.88/**54.71** (0.7/0.8) | 28.18/**38.44** (0.6/0.7) | 23.96/**35.31** (1.6/4.0) |
| | 74.86/**75.41** (2.0/2.0) | 69.46/**83.34** (3.1/2.0) | 74.12/**86.08** (2.0/1.5) | 60.61/**64.52** (1.5/1.3) | 38.47/**47.51** (5.5/3.7) |

Table 2: Mean test accuracy and average computational time per epoch (in the parenthesis) for the WebKB and WikipediaNetwork datasets with fixed $48/32/20\%$ splits. First row: GCN[7]/GCN-SCT. Second row: GCNII[7]/GCNII-SCT. (Unit:% for accuracy and $\times 10^{-2}$ second for computational time.)

# 7 CONCLUDING REMARKS

In this paper, we have established a geometric characterization of how ReLU and leaky ReLU affect the smoothness of the learned graph node representations in GCNs. We further study the dimension-wise normalized smoothness of the learned node features, showing that activation functions not only smooth node features but also can reduce or preserve the normalized smoothness of the features. Our theoretical findings inform the design of a simple and effective SCT for GCN. The proposed SCT can change the smoothness, in terms of both normalized and unnormalized smoothness, of the learned node features by GCN. Our work focuses on GCN with ReLU or leaky ReLU – the only two cases where the over-smoothing issue has been proved. Establishing theories of over-smoothing and controlling the smoothness of features for other activation functions is an interesting future direction.

ETHICS STATEMENT

Our paper focuses on developing new theoretical understandings of the smoothness of node features learned by graph convolutional networks. The paper is mainly theoretical. We do not see any potential ethical issues in our research; all experiments are carried out using existing benchmark settings and datasets.

REPRODUCIBILITY STATEMENT

We are committed to conducting reproducible research. For the theoretical proofs, we have provided detailed derivation to make sure it is easy to read by a broad audience. For the experiments part, we have submitted the code with detailed documentation to make it easy to reproduce the results reported in our paper.

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

# A  NOTATIONS

## A.1  SUMMARY OF NOTATIONS

| Notation | Meaning |
|---|---|
| $G = (V, E)$ | an undirected graph with vertex set $V = \{v_i\}_{i=1}^n$ and edge set $E$ |
| $\boldsymbol{A}$ | the adjacency matrix of the graph $G$ with entries $A_{ij} = \mathbf{1}_{(i,j) \in E}$ |
| $\boldsymbol{D}$ | the degree matrix of the graph $G$ with $D_{ii} = \sum_{j=1}^n A_{ij}$ |
| $\boldsymbol{G}$ | the (augmented) normalized adjacency matrix $\boldsymbol{G} := (\boldsymbol{D} + \boldsymbol{I})^{-\frac{1}{2}}(\boldsymbol{I} + \boldsymbol{A})(\boldsymbol{D} + \boldsymbol{I})^{-\frac{1}{2}}$ |
| $\boldsymbol{H}^l$ | node features at the $l^{th}$ layer of GCN where $\boldsymbol{H}^l := [(\boldsymbol{h}_1^l)^\top, \ldots, (\boldsymbol{h}_n^l)^\top]^\top \in \mathbb{R}^{d \times n}$ |
| $\sigma(\cdot)$ | the activation function, e.g. ReLU |
| $\sigma_a(\cdot)$ | leaky ReLU |
| $\boldsymbol{W}^l$ | the learnable weights of the $l^{th}$ layer of GCN |
| $\mathcal{M}, \mathcal{M}^\perp$ | the eigenspace corresponding to the eigenvalue 1 of matrix $\boldsymbol{G}$ and its orthogonal complement |
| $\lambda_i, \boldsymbol{e}_i$ | the $i^{th}$ largest eigenvalue of matrix $\boldsymbol{G}$ and the eigenvector corresponding to $\boldsymbol{e}_i$ |
| $\boldsymbol{H}_{\mathcal{M}}^l, \boldsymbol{H}_{\mathcal{M}^\perp}^l$ | the projections of $\boldsymbol{H}^l$ onto $\mathcal{M}$ and $\mathcal{M}^\perp$, respectively |
| $\| \cdot \|_{\mathcal{M}^\perp}, \| \cdot \|_E$ | the distance of node features $\boldsymbol{H}$ to the eigenspace $\mathcal{M}$ and the Dirichlet energy of $\boldsymbol{H}$ |
| $\lambda$ | the second largest magnitude of $\boldsymbol{G}$'s eigenvalues |
| $s_l$ | the largest singular value of $\boldsymbol{W}^l$ |
| $\boldsymbol{u}_i$ | the indicator vector of the $i^{th}$ connected component $V_i$ of the graph $G$ |
| $s(\cdot)$ | the normalized smoothness of a feature vector |
| $\boldsymbol{B}_\alpha^l$ | the smoothness control term at the $l^{th}$ layer |

Table 3: Some important notations used in our paper.

## A.2  DETAILS OF NOTATIONS

For two vectors $\boldsymbol{u} = (u_1, u_2, \ldots, u_d)$ and $\boldsymbol{v} = (v_1, v_2, \ldots, v_d)$, their inner product is defined as

$$\langle \boldsymbol{u}, \boldsymbol{v} \rangle = \sum_{i=1}^d u_i v_i,$$

their Hadamard product is defined as

$$\boldsymbol{u} \odot \boldsymbol{v} = (u_1 v_1, u_2 v_2, \ldots, u_d v_d),$$

and their Kronecker product is defined as

$$\boldsymbol{u} \otimes \boldsymbol{v} = \boldsymbol{u}\boldsymbol{v}^\top = \begin{pmatrix} u_1 v_1 & u_1 v_2 & \ldots & u_1 v_d \\ u_2 v_1 & u_2 v_2 & \ldots & u_2 v_d \\ \vdots & \vdots & \ddots & \vdots \\ u_d v_1 & u_d v_2 & \ldots & u_d v_d \end{pmatrix}.$$

The Kronecker product can be defined for two vectors of different lengths in a similar manner as above.

# B  PROOFS IN SECTION 3

First, we prove that the two smoothness notions used in [28; 5] are two equivalent seminorms, i.e. we prove Proposition 3.1 below.

*Proof of Proposition 3.1.* Notice that the matrix $\boldsymbol{H}$ can be decomposed as $\boldsymbol{H} = \sum_{i=1}^n \boldsymbol{H}\boldsymbol{e}_i\boldsymbol{e}_i^\top$ where each $\boldsymbol{e}_i$ is the eigenvector of $\boldsymbol{G}$ associated with eigenvalue $\lambda_i$ defined in subsection 2. This

indicates that

$$
\begin{aligned}
\boldsymbol{H}\tilde{\Delta} &= \boldsymbol{H}(\boldsymbol{I} - \boldsymbol{G}) \\
&= \sum_{i=1}^{n} \boldsymbol{H}\boldsymbol{e}_i\boldsymbol{e}_i^\top (\boldsymbol{I} - \boldsymbol{G}) \\
&= \sum_{i=1}^{n} (\boldsymbol{H}\boldsymbol{e}_i\boldsymbol{e}_i^\top - \boldsymbol{H}\boldsymbol{e}_i\boldsymbol{e}_i^\top \boldsymbol{G}) \\
&= \sum_{i=1}^{n} (\boldsymbol{H}\boldsymbol{e}_i\boldsymbol{e}_i^\top - \boldsymbol{H}\boldsymbol{e}_i(\lambda_i\boldsymbol{e}_i)^\top) \\
&= \sum_{i=1}^{n} (1 - \lambda_i)\boldsymbol{H}\boldsymbol{e}_i\boldsymbol{e}_i^\top \\
&= \sum_{i=m+1}^{n} (1 - \lambda_i)\boldsymbol{H}\boldsymbol{e}_i\boldsymbol{e}_i^\top.
\end{aligned}
$$

Then using the fact that $1 - \lambda_i \geq 0$ for each $i$, we obtain

$$
\begin{aligned}
\|\boldsymbol{H}\|_E^2 &= \mathrm{Trace}(\boldsymbol{H}\tilde{\Delta}\boldsymbol{H}^\top) \\
&= \mathrm{Trace}\Big( \sum_{i=m+1}^{n} (1 - \lambda_i)\boldsymbol{H}\boldsymbol{e}_i\boldsymbol{e}_i^\top (\sum_{j=1}^{n} \boldsymbol{H}\boldsymbol{e}_j\boldsymbol{e}_j^\top)^\top \Big) \\
&= \mathrm{Trace}\Big( \sum_{i=m+1}^{n}\sum_{j=1}^{n} (1 - \lambda_i)\boldsymbol{H}\boldsymbol{e}_i\boldsymbol{e}_i^\top \boldsymbol{e}_j\boldsymbol{e}_j^\top \boldsymbol{H}^\top \Big) \\
&= \mathrm{Trace}\Big( \sum_{i=m+1}^{n} (1 - \lambda_i)\boldsymbol{H}\boldsymbol{e}_i\boldsymbol{e}_i^\top \boldsymbol{e}_i\boldsymbol{e}_i^\top \boldsymbol{H}^\top \Big) \\
&= \mathrm{Trace}\Big( \sum_{i=m+1}^{n} \sqrt{1 - \lambda_i}\boldsymbol{H}\boldsymbol{e}_i\boldsymbol{e}_i^\top \boldsymbol{e}_i\boldsymbol{e}_i^\top \boldsymbol{H}^\top \sqrt{1 - \lambda_i} \Big) \\
&= \mathrm{Trace}\Big( \sum_{i=m+1}^{n} \sqrt{1 - \lambda_i}\boldsymbol{H}\boldsymbol{e}_i\boldsymbol{e}_i^\top ( \sum_{j=m+1}^{n} \sqrt{1 - \lambda_j}\boldsymbol{H}\boldsymbol{e}_j\boldsymbol{e}_j^\top)^\top \Big) \\
&= \Big\| \sum_{i=m+1}^{n} \sqrt{1 - \lambda_i}\boldsymbol{H}\boldsymbol{e}_i\boldsymbol{e}_i^\top \Big\|_F^2.
\end{aligned}
$$

That is,

$$
\|\boldsymbol{H}\|_E = \Big\| \sum_{i=m+1}^{n} \sqrt{1 - \lambda_i}\boldsymbol{H}\boldsymbol{e}_i\boldsymbol{e}_i^\top \Big\|_F.
$$

On the other hand, equation (3) implies

$$
\|\boldsymbol{H}\|_{\mathcal{M}^\perp} = \|\boldsymbol{H}_{\mathcal{M}^\perp}\|_F = \Big\| \sum_{i=m+1}^{n} \boldsymbol{H}\boldsymbol{e}_i\boldsymbol{e}_i^\top \Big\|_F.
$$

We first show that both $\|\boldsymbol{H}\|_{\mathcal{M}^\perp}$ and $\|\boldsymbol{H}\|_E$ are seminorms. Since $\|c\boldsymbol{H}\|_F = |c| \cdot \|\boldsymbol{H}\|_F$ for any $c \in \mathbb{R}$, we have $\|c\boldsymbol{H}\|_{\mathcal{M}^\perp} = |c| \cdot \|\boldsymbol{H}\|_{\mathcal{M}^\perp}$ and $\|c\boldsymbol{H}\|_E = |c| \cdot \|\boldsymbol{H}\|_E$. Moreover, for any two matrices $\boldsymbol{H}^1$ and $\boldsymbol{H}^2$ s.t. $\boldsymbol{H} = \boldsymbol{H}^1 + \boldsymbol{H}^2$, we have

$$
\sum_{i=m+1}^{n} \boldsymbol{H}^1\boldsymbol{e}_i\boldsymbol{e}_i^\top + \sum_{i=m+1}^{n} \boldsymbol{H}^2\boldsymbol{e}_i\boldsymbol{e}_i^\top = \sum_{i=m+1}^{n} \boldsymbol{H}\boldsymbol{e}_i\boldsymbol{e}_i^\top,
$$

$$
\sum_{i=m+1}^{n} \sqrt{1 - \lambda_i}\boldsymbol{H}^1\boldsymbol{e}_i\boldsymbol{e}_i^\top + \sum_{i=m+1}^{n} \sqrt{1 - \lambda_i}\boldsymbol{H}^2\boldsymbol{e}_i\boldsymbol{e}_i^\top = \sum_{i=m+1}^{n} \sqrt{1 - \lambda_i}\boldsymbol{H}\boldsymbol{e}_i\boldsymbol{e}_i^\top.
$$

Then the triangle inequality of Frobenius norm $\|\cdot\|_F$ implies that of $\|\boldsymbol{H}\|_{\mathcal{M}^\perp}$ and $\|\boldsymbol{H}\|_E$, respectively.

Now since $0 < 1 - \lambda_{m+1} \leq 1 - \lambda_i \leq 2$ for any $i = m+1, \ldots, n$, we may take $\alpha = \sqrt{1 - \lambda_{m+1}}$ and $\beta = \sqrt{2}$. Then we see that

$$
\begin{aligned}
\alpha\|\boldsymbol{H}\|_{\mathcal{M}^\perp} &= \left\|\alpha \sum_{i=m+1}^n \boldsymbol{H} e_i e_i^\top\right\|_F \\
&\leq \left\|\sum_{i=m+1}^n \sqrt{1-\lambda_i}\boldsymbol{H} e_i e_i^\top\right\|_F \\
&\leq \left\|\beta \sum_{i=m+1}^n \boldsymbol{H} e_i e_i^\top\right\|_F \\
&= \beta\|\boldsymbol{H}\|_{\mathcal{M}^\perp}.
\end{aligned}
$$

The result thus follows from $\|\boldsymbol{H}\|_E = \left\|\sum_{i=m+1}^n \sqrt{1-\lambda_i}\boldsymbol{H} e_i e_i^\top\right\|_F$. $\qquad\square$

### B.1 RELU

We present a crucial tool for characterizing how the activation function affects its input.

**Lemma B.1.** *Let $\boldsymbol{Z} \in \mathbb{R}^{d \times n}$, and let $\boldsymbol{Z}^+ = \max(\boldsymbol{Z}, 0)$ and $\boldsymbol{Z}^- = \max(-\boldsymbol{Z}, 0)$ be the (component-wise) positive and negative parts of $\boldsymbol{Z}$. Then 1) $\boldsymbol{Z}^+, \boldsymbol{Z}^-$ are (component-wise) nonnegative and $\boldsymbol{Z} = \boldsymbol{Z}^+ - \boldsymbol{Z}^-$ and 2) $\langle \boldsymbol{Z}^+, \boldsymbol{Z}^- \rangle_F = 0$.*

*Proof of Lemma B.1.* Notice that for any $a \in \mathbb{R}$, we have

$$
\max(a, 0) = \begin{cases} a & \text{if } a \geq 0 \\ 0 & \text{otherwise} \end{cases} \quad \text{and} \quad \max(-a, 0) = \begin{cases} 0 & \text{if } a \geq 0 \\ -a & \text{otherwise} \end{cases}
$$

where $a$ is any scalar. This implies that $a = \max(a, 0) - \max(-a, 0)$ and $\max(a, 0) \cdot \max(-a, 0) = 0$.

Let $Z_{ij}$ be the $(i, j)^{th}$ entry of $\boldsymbol{Z}$. Then $\boldsymbol{Z} = \boldsymbol{Z}^+ - \boldsymbol{Z}^-$ follows from $Z_{ij} = \max(Z_{ij}, 0) - \max(-Z_{ij}, 0)$. Also, one can deduce that

$$
\begin{aligned}
\langle \boldsymbol{Z}^+, \boldsymbol{Z}^- \rangle_F &= \text{Trace}((\boldsymbol{Z}^+)^\top \boldsymbol{Z}^-) \\
&= \sum_{i=1}^d \sum_{j=1}^j \max(Z_{ij}, 0) \max(-Z_{ij}, 0) \\
&= 0.
\end{aligned}
$$

$\qquad\square$

Before proving Proposition 3.2, we notice the following relation between $\boldsymbol{Z}$ and $\boldsymbol{H}$.

**Lemma B.2.** *Given $\boldsymbol{Z} \in \mathbb{R}^{d \times n}$, let $\boldsymbol{H} = \sigma(\boldsymbol{Z})$ with $\sigma$ being ReLU, then $\boldsymbol{H}$ lies on the high-dimensional sphere, in $\|\cdot\|_F$ norm, that is centered at $\boldsymbol{Z}/2$ and with radius $\|\boldsymbol{Z}/2\|_F$. That is, $\boldsymbol{H}$ and $\boldsymbol{Z}$ satisfy the following equation*

$$
\left\|\boldsymbol{H} - \frac{\boldsymbol{Z}}{2}\right\|_F^2 = \left\|\frac{\boldsymbol{Z}}{2}\right\|_F^2. \tag{8}
$$

*Proof of Lemma B.2.* We observe that $\boldsymbol{H} = \sigma(\boldsymbol{Z}) = \max(\boldsymbol{Z}, 0) = \boldsymbol{Z}^+$ is the positive part of $\boldsymbol{Z}$. Then we have

$$
\begin{aligned}
\langle \boldsymbol{H}, \boldsymbol{Z} \rangle_F &= \langle \boldsymbol{H}, \boldsymbol{Z}^+ - \boldsymbol{Z}^- \rangle_F \\
&= \langle \boldsymbol{H}, \boldsymbol{Z}^+ \rangle_F - \langle \boldsymbol{H}, \boldsymbol{Z}^- \rangle_F \\
&= \langle \boldsymbol{H}, \boldsymbol{H} \rangle_F,
\end{aligned}
$$

where we have used $\boldsymbol{Z} = \boldsymbol{Z}^+ - \boldsymbol{Z}^-$ and $\langle \boldsymbol{H}, \boldsymbol{Z}^- \rangle_F = \langle \boldsymbol{Z}^+, \boldsymbol{Z}^- \rangle_F = 0$ from Lemma B.1.

Therefore, one can deduce the desired result as follows:

$$\langle \boldsymbol{H}, \boldsymbol{H} \rangle_F - \langle \boldsymbol{H}, \boldsymbol{Z} \rangle_F = 0$$

$$\Rightarrow \|\boldsymbol{H}\|_F^2 - 2\left\langle \boldsymbol{H}, \frac{\boldsymbol{Z}}{2} \right\rangle_F + \left\|\frac{\boldsymbol{Z}}{2}\right\|_F^2 = \left\|\frac{\boldsymbol{Z}}{2}\right\|_F^2$$

$$\Rightarrow \left\|\boldsymbol{H} - \frac{\boldsymbol{Z}}{2}\right\|_F^2 = \left\|\frac{\boldsymbol{Z}}{2}\right\|_F^2.$$

$\square$

Applying $\|\boldsymbol{H}\|_F^2 = \|\boldsymbol{H}_{\mathcal{M}} + \boldsymbol{H}_{\mathcal{M}^\perp}\|_F^2 = \|\boldsymbol{H}_{\mathcal{M}}\|_F^2 + \|\boldsymbol{H}_{\mathcal{M}^\perp}\|_F^2$, to both $\frac{\boldsymbol{Z}}{2}$ and $\boldsymbol{H} - \frac{\boldsymbol{Z}}{2}$, we obtain

$$\left\|\frac{\boldsymbol{Z}}{2}\right\|_F^2 = \left\|\frac{\boldsymbol{Z}_{\mathcal{M}^\perp}}{2}\right\|_F^2 + \left\|\frac{\boldsymbol{Z}_{\mathcal{M}}}{2}\right\|_F^2,$$

and

$$\left\|\boldsymbol{H} - \frac{\boldsymbol{Z}}{2}\right\|_F^2 = \left\|\boldsymbol{H}_{\mathcal{M}^\perp} - \frac{\boldsymbol{Z}_{\mathcal{M}^\perp}}{2}\right\|_F^2 + \left\|\boldsymbol{H}_{\mathcal{M}} - \frac{\boldsymbol{Z}_{\mathcal{M}}}{2}\right\|_F^2.$$

Then equation (8) becomes

$$\left\|\frac{\boldsymbol{Z}_{\mathcal{M}^\perp}}{2}\right\|_F^2 - \left\|\boldsymbol{H}_{\mathcal{M}^\perp} - \frac{\boldsymbol{Z}_{\mathcal{M}^\perp}}{2}\right\|_F^2 = \left\|\boldsymbol{H}_{\mathcal{M}} - \frac{\boldsymbol{Z}_{\mathcal{M}}}{2}\right\|_F^2 - \left\|\frac{\boldsymbol{Z}_{\mathcal{M}}}{2}\right\|_F^2 \tag{9}$$

By direct calculation, we have

$$\left\|\boldsymbol{H}_{\mathcal{M}} - \frac{\boldsymbol{Z}_{\mathcal{M}}}{2}\right\|_F^2 - \left\|\frac{\boldsymbol{Z}_{\mathcal{M}}}{2}\right\|_F^2$$

$$= \langle \boldsymbol{H}_{\mathcal{M}}, \boldsymbol{H}_{\mathcal{M}} \rangle_F - 2\left\langle \boldsymbol{H}_{\mathcal{M}}, \frac{\boldsymbol{Z}_{\mathcal{M}}}{2} \right\rangle_F \tag{10}$$

$$= \langle \boldsymbol{H}_{\mathcal{M}}, \boldsymbol{H}_{\mathcal{M}} - \boldsymbol{Z}_{\mathcal{M}} \rangle_F.$$

Combining equation (9) and equation (10), we obtain the following result:

**Lemma B.3.** *For any $\boldsymbol{Z} = \boldsymbol{Z}_{\mathcal{M}} + \boldsymbol{Z}_{\mathcal{M}^\perp}$, let $\boldsymbol{H} = \sigma(\boldsymbol{Z}) = \boldsymbol{H}_{\mathcal{M}} + \boldsymbol{H}_{\mathcal{M}^\perp}$, then*

$$\left\|\frac{\boldsymbol{Z}_{\mathcal{M}^\perp}}{2}\right\|_F^2 - \left\|\boldsymbol{H}_{\mathcal{M}^\perp} - \frac{\boldsymbol{Z}_{\mathcal{M}^\perp}}{2}\right\|_F^2 = \langle \boldsymbol{Z}_{\mathcal{M}}^+, \boldsymbol{Z}_{\mathcal{M}}^- \rangle_F.$$

*where $\boldsymbol{Z}_{\mathcal{M}}^+ = \sum_{i=1}^m \boldsymbol{Z}^+ \boldsymbol{e}_i \boldsymbol{e}_i^\top, \boldsymbol{Z}_{\mathcal{M}}^- = \sum_{i=1}^m \boldsymbol{Z}^- \boldsymbol{e}_i \boldsymbol{e}_i^\top.$*

*Proof of Lemma B.3.* Recall that $\boldsymbol{H} = \sigma(\boldsymbol{Z}) = \max(\boldsymbol{Z}, 0) = \boldsymbol{Z}^+$. Also, $\boldsymbol{Z} = \boldsymbol{Z}^+ - \boldsymbol{Z}^-$ implies $\boldsymbol{Z}_{\mathcal{M}} = \boldsymbol{Z}_{\mathcal{M}}^+ - \boldsymbol{Z}_{\mathcal{M}}^- = \boldsymbol{H}_{\mathcal{M}}^+ - \boldsymbol{Z}_{\mathcal{M}}^-$. Therefore, we see that

$$\langle \boldsymbol{H}_{\mathcal{M}}, \boldsymbol{H}_{\mathcal{M}} - \boldsymbol{Z}_{\mathcal{M}} \rangle_F = \langle \boldsymbol{Z}_{\mathcal{M}}^+, \boldsymbol{Z}_{\mathcal{M}}^- \rangle_F.$$

$\square$

By using the fact that $\langle \boldsymbol{Z}_{\mathcal{M}}^+, \boldsymbol{Z}_{\mathcal{M}}^- \rangle_F \geq 0$ in Lemma B.3, we reveal a geometric relation between $\boldsymbol{Z}$ and $\boldsymbol{H}$ mentioned in Proposition 3.2.

*Proof of Proposition 3.2.* Since $\boldsymbol{Z}^+, \boldsymbol{Z}^- \geq 0$ are nonnegative and all the eigenvectors $\boldsymbol{e}_i$ are also nonnegative, we see that $\boldsymbol{Z}_{\mathcal{M}}^+ = \sum_{i=1}^m \boldsymbol{Z}^+ \boldsymbol{e}_i \boldsymbol{e}_i^\top$ and $\boldsymbol{Z}_{\mathcal{M}}^- = \sum_{i=1}^m \boldsymbol{Z}^- \boldsymbol{e}_i \boldsymbol{e}_i^\top$ are nonnegative. This indicates that

$$\langle \boldsymbol{Z}_{\mathcal{M}}^+, \boldsymbol{Z}_{\mathcal{M}}^- \rangle_F = \mathrm{Trace}\left(\boldsymbol{Z}_{\mathcal{M}}^+ (\boldsymbol{Z}_{\mathcal{M}}^-)^\top\right) \geq 0.$$

Then according to Lemma B.3, we obtain

$$\left\|\frac{\boldsymbol{Z}_{\mathcal{M}^\perp}}{2}\right\|_F^2 - \left\|\boldsymbol{H}_{\mathcal{M}^\perp} - \frac{\boldsymbol{Z}_{\mathcal{M}^\perp}}{2}\right\|_F^2 = \langle \boldsymbol{Z}_{\mathcal{M}}^+, \boldsymbol{Z}_{\mathcal{M}}^- \rangle_F \geq 0.$$

So we have

$$\left\|\boldsymbol{H}_{\mathcal{M}^\perp} - \frac{\boldsymbol{Z}_{\mathcal{M}^\perp}}{2}\right\|_F = \sqrt{\left\|\frac{\boldsymbol{Z}_{\mathcal{M}^\perp}}{2}\right\|_F^2 - \langle \boldsymbol{Z}_{\mathcal{M}}^+, \boldsymbol{Z}_{\mathcal{M}}^- \rangle_F}$$

$$= \sqrt{\left\|\frac{\boldsymbol{Z}_{\mathcal{M}^\perp}}{2}\right\|_F^2 - \langle \boldsymbol{H}_{\mathcal{M}}, \boldsymbol{H}_{\mathcal{M}} - \boldsymbol{Z}_{\mathcal{M}} \rangle_F},$$

which shows that $\boldsymbol{H}_{\mathcal{M}^\perp}$ lies on the high-dimensional sphere that we have claimed. Furthermore, we conclude that

$$0 \leq \left\| \boldsymbol{H}_{\mathcal{M}^\perp} - \frac{\boldsymbol{Z}_{\mathcal{M}^\perp}}{2} \right\|_F \leq \left\| \frac{\boldsymbol{Z}_{\mathcal{M}^\perp}}{2} \right\|_F. \tag{11}$$

This demonstrates that $\boldsymbol{H}_{\mathcal{M}^\perp}$ lies on the high-dimensional sphere we have stated.

Since the sphere $\left\| \boldsymbol{H}_{\mathcal{M}^\perp} - \frac{\boldsymbol{Z}_{\mathcal{M}^\perp}}{2} \right\|_F^2 = \left\| \frac{\boldsymbol{Z}_{\mathcal{M}^\perp}}{2} \right\|_F^2$ passes through the origin, the distance of any $\boldsymbol{H}_{\mathcal{M}^\perp}$ to the origin must be no greater than the diameter of this sphere, i.e. $\|\boldsymbol{H}_{\mathcal{M}^\perp}\|_F \leq \|\boldsymbol{Z}_{\mathcal{M}^\perp}\|_F$. Also, this can be derived from

$$\|\boldsymbol{H}_{\mathcal{M}^\perp}\|_F - \left\| \frac{\boldsymbol{Z}_{\mathcal{M}^\perp}}{2} \right\|_F \leq \left\| \boldsymbol{H}_{\mathcal{M}^\perp} - \frac{\boldsymbol{Z}_{\mathcal{M}^\perp}}{2} \right\|_F \leq \left\| \frac{\boldsymbol{Z}_{\mathcal{M}^\perp}}{2} \right\|_F.$$

One can see that the maximal smoothness $\|\boldsymbol{H}_{\mathcal{M}^\perp}\|_F = \|\boldsymbol{Z}_{\mathcal{M}^\perp}\|_F$ is attained when $\boldsymbol{H}_{\mathcal{M}^\perp} = \boldsymbol{Z}_{\mathcal{M}^\perp}$, the intersection of the surface and the line passing through the center and the origin.

After all, we complete the proof by using the fact that $\|\boldsymbol{Z}_{\mathcal{M}^\perp}\|_F = \|\boldsymbol{Z}\|_{\mathcal{M}^\perp}$ for any matrix $\boldsymbol{Z}$, which implies $\|\boldsymbol{H}\|_{\mathcal{M}^\perp} = \|\boldsymbol{H}_{\mathcal{M}^\perp}\|_F \leq \|\boldsymbol{Z}_{\mathcal{M}^\perp}\|_F = \|\boldsymbol{Z}\|_{\mathcal{M}^\perp}$.

$\square$

## B.2 LEAKY RELU

For the leaky ReLU activation function, we have

**Lemma B.4.** *If $\boldsymbol{H} = \sigma_a(\boldsymbol{Z})$ with $\sigma_a$ being leaky ReLU, then $\boldsymbol{H}$ lies on the high-dimensional sphere centered at $(1+a)\boldsymbol{Z}/2$ with radius $\|(1-a)\boldsymbol{Z}/2\|_F$.*

*Proof of Lemma B.4.* By writing $\boldsymbol{Z}$ into the sum of the positive part $\boldsymbol{Z}^+$ and negative part $\boldsymbol{Z}^-$, we see that

$$\boldsymbol{H} = \sigma_a(\boldsymbol{Z}) = \boldsymbol{Z}^+ - a\boldsymbol{Z}^-.$$

Then we have $\boldsymbol{H} - \boldsymbol{Z} = (1-a)\boldsymbol{Z}^-$ and $\boldsymbol{H} - a\boldsymbol{Z} = (1-a)\boldsymbol{Z}^+$. Using $\langle \boldsymbol{Z}^-, \boldsymbol{Z}^+ \rangle_F = 0$, we obtain the following

$$\langle \boldsymbol{H} - \boldsymbol{Z}, \boldsymbol{H} - a\boldsymbol{Z} \rangle_F = 0$$

$$\Rightarrow \|\boldsymbol{H}\|_F^2 - 2\left\langle \boldsymbol{H}, \frac{(1+a)\boldsymbol{Z}}{2} \right\rangle_F + a\|\boldsymbol{Z}\|_F^2 = 0$$

$$\Rightarrow \|\boldsymbol{H}\|_F^2 - 2\left\langle \boldsymbol{H}, \frac{(1+a)\boldsymbol{Z}}{2} \right\rangle_F = -a\|\boldsymbol{Z}\|_F^2$$

$$\Rightarrow \left\| \boldsymbol{H} - \frac{(1+a)}{2}\boldsymbol{Z} \right\|_F^2 = \left\| \frac{(1+a)}{2}\boldsymbol{Z} \right\|_F^2 - a\|\boldsymbol{Z}\|_F^2 = \left\| \frac{(1-a)}{2}\boldsymbol{Z} \right\|_F^2.$$

$\square$

Moreover, we notice that

**Lemma B.5.** *For any $\boldsymbol{Z} = \boldsymbol{Z}_\mathcal{M} + \boldsymbol{Z}_{\mathcal{M}^\perp}$, let $\boldsymbol{H} = \sigma_a(\boldsymbol{Z}) = \boldsymbol{H}_\mathcal{M} + \boldsymbol{H}_{\mathcal{M}^\perp}$, then*

$$\left\| \frac{(1-a)}{2}\boldsymbol{Z}_{\mathcal{M}^\perp} \right\|_F^2 - \left\| \boldsymbol{H}_{\mathcal{M}^\perp} - \frac{(1+a)}{2}\boldsymbol{Z}_{\mathcal{M}^\perp} \right\|_F^2 = (1-a)^2 \langle \boldsymbol{Z}_\mathcal{M}^+, \boldsymbol{Z}_\mathcal{M}^- \rangle_F$$

*Proof of Lemma B.5.* Similar to the proof of Lemma B.3, the orthogonal decomposition implies that

$$\left\| \frac{(1-a)}{2}\boldsymbol{Z}_{\mathcal{M}^\perp} \right\|_F^2 - \left\| \boldsymbol{H}_{\mathcal{M}^\perp} - \frac{(1+a)}{2}\boldsymbol{Z}_{\mathcal{M}^\perp} \right\|_F^2$$

$$= \left\| \boldsymbol{H}_\mathcal{M} - \frac{(1+a)}{2}\boldsymbol{Z}_\mathcal{M} \right\|_F^2 - \left\| \frac{(1-a)}{2}\boldsymbol{Z}_\mathcal{M} \right\|_F^2$$

$$= \langle \boldsymbol{H}_\mathcal{M} - \boldsymbol{Z}_\mathcal{M}, \boldsymbol{H}_\mathcal{M} - a\boldsymbol{Z}_\mathcal{M} \rangle_F$$

$$= \langle (1-a)\boldsymbol{Z}_\mathcal{M}^-, (1-a)\boldsymbol{Z}_\mathcal{M}^+ \rangle_F$$

$$= (1-a)^2 \langle \boldsymbol{Z}_\mathcal{M}^-, \boldsymbol{Z}_\mathcal{M}^+ \rangle_F.$$

$\square$

*Proof of Proposition 3.3.* Similar to the proof of Proposition 3.2, we apply the fact $\langle \boldsymbol{Z}_{\mathcal{M}}^-, \boldsymbol{Z}_{\mathcal{M}}^+ \rangle_F \geq 0$ to Lemma B.5 and hence obtain the geometric condition as follows:

$$\left\| \boldsymbol{H}_{\mathcal{M}^\perp} - \frac{(1+a)}{2} \boldsymbol{Z}_{\mathcal{M}^\perp} \right\|_F = \sqrt{\left\| \frac{(1-a)}{2} \boldsymbol{Z}_{\mathcal{M}^\perp} \right\|_F^2 - \langle \boldsymbol{H}_{\mathcal{M}} - \boldsymbol{Z}_{\mathcal{M}}, \boldsymbol{H}_{\mathcal{M}} - a\boldsymbol{Z}_{\mathcal{M}} \rangle_F}.$$

Then we have the following inequality:

$$0 \leq \left\| \boldsymbol{H}_{\mathcal{M}^\perp} - \frac{(1+a)}{2} \boldsymbol{Z}_{\mathcal{M}^\perp} \right\|_F \leq \left\| \frac{(1-a)}{2} \boldsymbol{Z}_{\mathcal{M}^\perp} \right\|_F.$$

Moreover, we deduce that

$$\left| \| \boldsymbol{H}_{\mathcal{M}^\perp} \|_F - \left\| \frac{(1+a)}{2} \boldsymbol{Z}_{\mathcal{M}^\perp} \right\|_F \right| \leq \left\| \boldsymbol{H}_{\mathcal{M}^\perp} - \frac{(1+a)}{2} \boldsymbol{Z}_{\mathcal{M}^\perp} \right\|_F$$

$$\leq \left\| \frac{(1-a)}{2} \boldsymbol{Z}_{\mathcal{M}^\perp} \right\|_F.$$

and hence

$$-\left\| \frac{(1-a)}{2} \boldsymbol{Z}_{\mathcal{M}^\perp} \right\|_F \leq \| \boldsymbol{H}_{\mathcal{M}^\perp} \|_F - \left\| \frac{(1+a)}{2} \boldsymbol{Z}_{\mathcal{M}^\perp} \right\|_F$$

$$\leq \left\| \frac{(1-a)}{2} \boldsymbol{Z}_{\mathcal{M}^\perp} \right\|_F.$$

Therefore, we obtain $a\|\boldsymbol{Z}_{\mathcal{M}^\perp}\|_F \leq \|\boldsymbol{H}_{\mathcal{M}^\perp}\|_F \leq \|\boldsymbol{Z}_{\mathcal{M}^\perp}\|_F$. (Remark that $\boldsymbol{H}_{\mathcal{M}^\perp}$ achieves its maximal norm when it is equal to $\boldsymbol{Z}_{\mathcal{M}^\perp}$, the intersection of the surface and the line passing through the center and the origin. )

By using the fact that $\|\boldsymbol{Z}_{\mathcal{M}^\perp}\|_F = \|\boldsymbol{Z}\|_{\mathcal{M}^\perp}$ for any matrix $\boldsymbol{Z}$, we conclude that $a\|\boldsymbol{Z}\|_{\mathcal{M}^\perp} \leq \|\boldsymbol{H}\|_{\mathcal{M}^\perp} \leq \|\boldsymbol{Z}\|_{\mathcal{M}^\perp}$. $\square$

## C    PROOFS IN SECTION 4

Throughout this section, we assume that $\boldsymbol{z}_{\mathcal{M}^\perp} \neq \boldsymbol{0}$.

*Proof of Proposition 4.2.* Recall that $\boldsymbol{e} = \tilde{\boldsymbol{D}}^{\frac{1}{2}} \boldsymbol{u}_n / c$ has only positive entries where $\tilde{\boldsymbol{D}}$ is the augmented degree matrix and $\boldsymbol{u}_n = [1, \dots, 1]^\top \in \mathbb{R}^n$ is the vector of ones and $c = \|\tilde{\boldsymbol{D}}^{\frac{1}{2}} \boldsymbol{u}_n\|$. Let $d_i$ be the $i^{th}$ diagonal entry of $\tilde{\boldsymbol{D}}$. Then we have

$$\boldsymbol{e} = [\sqrt{d_1}/c, \sqrt{d_2}/c, \dots, \sqrt{d_n}/c]^\top$$

and $c = \sqrt{\sum_{i=1}^n d_i}$.

Note that $\boldsymbol{z}(\alpha) = \boldsymbol{z} - \alpha \boldsymbol{e} = \boldsymbol{z} - \frac{\alpha}{c} \tilde{\boldsymbol{D}}^{\frac{1}{2}} \boldsymbol{u}_n = \tilde{\boldsymbol{D}}^{\frac{1}{2}} (\tilde{\boldsymbol{D}}^{-\frac{1}{2}} \boldsymbol{z} - \frac{\alpha}{c} \boldsymbol{u}_n) = \tilde{\boldsymbol{D}}^{\frac{1}{2}} (\boldsymbol{x} - \frac{\alpha}{c} \boldsymbol{u}_n)$, where we assume $\boldsymbol{x} := \tilde{\boldsymbol{D}}^{-\frac{1}{2}} \boldsymbol{z}$. Then we observe that when $\sigma$ is the ReLU activation function,

$$\boldsymbol{h}(\alpha) = \sigma(\boldsymbol{z}(\alpha))$$

$$= \sigma\left( \tilde{\boldsymbol{D}}^{\frac{1}{2}} \left( \boldsymbol{x} - \frac{\alpha}{c} \boldsymbol{u}_n \right) \right)$$

$$= \tilde{\boldsymbol{D}}^{\frac{1}{2}} \sigma\left( \boldsymbol{x} - \frac{\alpha}{c} \boldsymbol{u}_n \right),$$

and hence

$$\langle \boldsymbol{h}(\alpha), \boldsymbol{e} \rangle = \left\langle \tilde{\boldsymbol{D}}^{\frac{1}{2}} \sigma\left( \boldsymbol{x} - \frac{\alpha}{c} \boldsymbol{u}_n \right), \boldsymbol{e} \right\rangle$$

$$= \left\langle \sigma\left( \boldsymbol{x} - \frac{\alpha}{c} \boldsymbol{u}_n \right), \tilde{\boldsymbol{D}}^{\frac{1}{2}} \boldsymbol{e} \right\rangle$$

$$= \left\langle \sigma\left( \boldsymbol{x} - \frac{\alpha}{c} \boldsymbol{u}_n \right), \tilde{\boldsymbol{D}} \boldsymbol{u}_n \right\rangle.$$

We may now assume $\boldsymbol{x} = [x_1, \ldots, x_n]^\top$ is well-ordered s.t. $x_1 \geq x_2 \geq \ldots \geq x_n$. Indeed, there is a collection of indices $\{k_1, \ldots, k_l\}$ s.t.

$$x_1 = \ldots, x_{k_1} \text{ and } x_{k_1} > x_{k_1+1},$$
$$x_{k_{j-1}+1} = \ldots = x_{k_j} \text{ and } x_{k_j} > x_{k_j+1} \text{ for any } j = 2, \ldots, l-1,$$
$$x_{k_{l-1}+1} = \ldots = x_{k_l} \text{ and } k_l = n.$$

That is, $x_1 = x_2 = \ldots = x_{k_1} > x_{k_1+1} = \ldots = x_{k_2} > x_{k_2+1} = \ldots = x_{k_3} > x_{k_3+1} \ldots$

We first restrict the domain of $\alpha$ s.t. $\boldsymbol{h}(\alpha) \neq 0$. Note that we have

$$\boldsymbol{h}(\alpha) = 0$$
$$\Leftrightarrow \sigma\left(\boldsymbol{x} - \frac{\alpha}{c}\boldsymbol{u}_n\right) = 0$$
$$\Leftrightarrow x_i - \frac{\alpha}{c} \leq 0 \text{ for } i = 1, \ldots, n$$
$$\Leftrightarrow x_1 - \frac{\alpha}{c} \leq 0$$
$$\Leftrightarrow \alpha \geq cx_1.$$

So we will study the smoothness $s(\boldsymbol{h}(\alpha))$ when $\alpha < cx_1$.

Let $\epsilon > 0$ be a positive number and consider $\alpha = c(x_1 - \epsilon)$. When $\epsilon \leq x_1 - x_{k_1+1} = x_1 - x_{k_2}$, we see that

$$\boldsymbol{x} - \frac{\alpha}{c}\boldsymbol{u}_n = [\epsilon, \ldots, \epsilon, \epsilon - (x_1 - x_{k_1+1}), \ldots, \epsilon - (x_1 - x_n)]^\top,$$

where only the first $k_1$ entries are positive since $x_1 - x_i \geq \epsilon$ for any $i \geq k_1 + 1$. Therefore,

$$\boldsymbol{h}(\alpha) = \tilde{\boldsymbol{D}}^{\frac{1}{2}}\sigma\left(\boldsymbol{x} - \frac{\alpha}{c}\boldsymbol{u}_n\right)$$
$$= \tilde{\boldsymbol{D}}^{\frac{1}{2}}[\epsilon, \ldots, \epsilon, 0, \ldots, 0]^\top$$
$$= [\epsilon\sqrt{d_1}, \ldots, \epsilon\sqrt{d_{k_1}}, 0, \ldots, 0]^\top.$$

and hence we can compute that

$$\|\boldsymbol{h}(\alpha)\| = \epsilon\sqrt{\sum_{i=1}^{k_1} d_i}.$$

Also, we have

$$\|\boldsymbol{h}(\alpha)\|_{\mathcal{M}} = |\langle \boldsymbol{h}(\alpha), \boldsymbol{e}\rangle|$$
$$= [\epsilon\sqrt{d_1}, \ldots, \epsilon\sqrt{d_{k_1}}, 0, \ldots, 0]^\top [\sqrt{d_1}/c, \sqrt{d_2}/c, \ldots, \sqrt{d_n}/c]$$
$$= \frac{\epsilon}{c}\sum_{i=1}^{k_1} d_i.$$

Then we obtain the smoothness $s(\boldsymbol{h}(\alpha))$ as follows:

$$s(\boldsymbol{h}(\alpha)) = \frac{\|\boldsymbol{h}(\alpha)\|_{\mathcal{M}}}{\|\boldsymbol{h}(\alpha)\|} = \frac{\frac{\epsilon}{c}\sum_{i=1}^{k_1} d_i}{\epsilon\sqrt{\sum_{i=1}^{k_1} d_i}} = \frac{\sqrt{\sum_{i=1}^{k_1} d_i}}{c} = \frac{K_1}{c} < 1,$$

where we denote $\sqrt{\sum_{i=1}^{k_1} d_i}$ by $K_1$. Similarly, we may denote $\sqrt{\sum_{i=k_{j-1}+1}^{k_j} d_i}$ by $K_j$ for $j = 2, \ldots, l$.

Now we are going to show that the smoothness $s(\boldsymbol{h}(\alpha))$ is increasing as $\alpha$ gets smaller whenever $\alpha < cx_1$, which further implies $\frac{K_1}{c}$ is the minimum of the smoothness $s(\boldsymbol{h}(\alpha))$. Remember that we are considering $\alpha = c(x_1 - \epsilon)$ and we have studied the case when $0 < \epsilon \leq x_1 - x_{k_1+1} = x_1 - x_{k_2}$.

Let $\delta_j := x_1 - x_{k_j}$ for $1 \le j \le l$. Clearly, we have $\delta_1 = 0$ and $\delta_j < \delta_{j+1}$ for $1 \le j \le l-1$. Fix a $j' \in \{2, \ldots, l-1\}$. We see that when $\delta_{j'} < \epsilon \le x_1 - x_{k_{j'}+1}$,

$$
\boldsymbol{x} - \frac{\alpha}{c}\boldsymbol{u}_n
$$
$$
= \Big[\epsilon - \delta_1, \ldots, \epsilon - \delta_1, \epsilon - \delta_2, \ldots, \epsilon - \delta_2, \epsilon - \delta_3, \ldots, \epsilon - \delta_{j'},
$$
$$
\epsilon - (x_1 - x_{k_{j'}+1}), \ldots, \epsilon - (x_1 - x_n)\Big]^\top,
$$

where we have $\epsilon - \delta_j > 0$ for any $2 \le j \le j'$ and $\epsilon - (x_1 - x_i) \le 0$ for any $i \ge k_{j'}+1$. Consequently,

$$
\boldsymbol{h}(\alpha) = \tilde{\boldsymbol{D}}^{\frac{1}{2}}\sigma(\boldsymbol{x} - \frac{\alpha}{c}\boldsymbol{u}_n)
$$
$$
= [(\epsilon - \delta_1)\sqrt{d_1}, \ldots, (\epsilon - \delta_1)\sqrt{d_{k_1}}, (\epsilon - \delta_2)\sqrt{d_{k_1+1}}, \ldots, (\epsilon - \delta_2)\sqrt{d_{k_2}},
$$
$$
(\epsilon - \delta_3)\sqrt{d_{k_2+1}}, \ldots, (\epsilon - \delta_{j'})\sqrt{d_{k_{j'}}}, 0, \ldots, 0]^\top.
$$

Then we can compute

$$
\|\boldsymbol{h}(\alpha)\| = \sqrt{\sum_{j=1}^{j'}\sum_{i=k_{j-1}+1}^{k_j} d_i(\epsilon - \delta_j)^2} = \sqrt{\sum_{j=1}^{j'} K_j^2(\epsilon - \delta_j)^2},
$$

where we set $k_0 := 0$ for simplicity and $K_j = \sqrt{\sum_{i=k_{j-1}+1}^{k_j} d_i}$ for $j = 1, \ldots, j'$. Also, we have

$$
\|\boldsymbol{h}(\alpha)\|_{\mathcal{M}} = |\langle \boldsymbol{h}(\alpha), \boldsymbol{e}\rangle| = \sum_{j=1}^{j'}\sum_{i=k_{j-1}+1}^{k_j} \frac{d_i(\epsilon - \delta_j)}{c} = \frac{1}{c}\sum_{j=1}^{j'} K_j^2(\epsilon - \delta_j).
$$

A careful calculation shows that $\frac{\partial}{\partial\epsilon}s(\boldsymbol{h}(\alpha)) > 0$ whenever $\delta_{j'} < \epsilon \le x_1 - x_{k_{j'}+1}$ which implies that $s(\boldsymbol{h}(\alpha))$ is increasing as $\epsilon$ increases. Indeed, we have

$$
\frac{\partial}{\partial\epsilon}s(\boldsymbol{h}(\alpha))
$$
$$
= \frac{\partial}{\partial\epsilon}\left(\frac{\sum_{j=1}^{j'} K_j^2(\epsilon - \delta_j)}{c\sqrt{\sum_{j=1}^{j'} K_j^2(\epsilon - \delta_j)^2}}\right)
$$
$$
= \frac{\left(\frac{\partial}{\partial\epsilon}\sum_{j=1}^{j'} K_j^2(\epsilon - \delta_j)\right)\sqrt{\sum_{j=1}^{j'} K_j^2(\epsilon - \delta_j)^2} - \sum_{j=1}^{j'} K_j^2(\epsilon - \delta_j)\left(\frac{\partial}{\partial\epsilon}\sqrt{\sum_{j=1}^{j'} K_j^2(\epsilon - \delta_j)^2}\right)}{c\sum_{j=1}^{j'} K_j^2(\epsilon - \delta_j)^2}
$$
$$
= \frac{\left(\sum_{j=1}^{j'} K_j^2\right)\sqrt{\sum_{j=1}^{j'} K_j^2(\epsilon - \delta_j)^2} - \sum_{j=1}^{j'} K_j^2(\epsilon - \delta_j)\left(\frac{\frac{\partial}{\partial\epsilon}\sum_{j=1}^{j'} K_j^2(\epsilon - \delta_j)^2}{2\sqrt{\sum_{j=1}^{j'} K_j^2(\epsilon - \delta_j)^2}}\right)}{c\sum_{j=1}^{j'} K_j^2(\epsilon - \delta_j)^2}
$$
$$
= \frac{\left(\sum_{j=1}^{j'} K_j^2\right)\sum_{j=1}^{j'} K_j^2(\epsilon - \delta_j)^2 - \sum_{j=1}^{j'} K_j^2(\epsilon - \delta_j)\left(\sum_{j=1}^{j'} K_j^2(\epsilon - \delta_j)\right)}{c\sum_{j=1}^{j'} K_j^2(\epsilon - \delta_j)^2\sqrt{\sum_{j=1}^{j'} K_j^2(\epsilon - \delta_j)^2}}.
$$

Then to show that $\frac{\partial}{\partial\epsilon}s(\boldsymbol{h}(\alpha)) > 0$, it suffices to show that the numerator is positive, i.e.

$$
\left(\sum_{j=1}^{j'} K_j^2\right)\sum_{j=1}^{j'} K_j^2(\epsilon - \delta_j)^2 - \left(\sum_{j=1}^{j'} K_j^2(\epsilon - \delta_j)\right)^2 > 0,
$$

since the denominator $c\sum_{j=1}^{j'} K_j^2(\epsilon - \delta_j)^2\sqrt{\sum_{j=1}^{j'} K_j^2(\epsilon - \delta_j)^2} > 0$ is always positive. In fact, this follows from the Cauchy inequality $\|\boldsymbol{v}\|\|\boldsymbol{u}\| \ge \langle \boldsymbol{v}, \boldsymbol{u}\rangle$, where we set

$$
\boldsymbol{v} := [K_1, K_2, \ldots, K_{J'}]^\top,
$$
$$
\boldsymbol{u} := [K_1(\epsilon - \delta_1), K_2(\epsilon - \delta_2), \ldots, K_{j'}(\epsilon - \delta_{j'})]^\top.
$$

Moreover, equality happens only when $v$ is parallel to $u$. This is, however, impossible since $\epsilon - \delta_j > \epsilon - \delta_{j+1}$ for any $j = 1, \ldots, j' - 1$ and each $K_j$ is positive.

So we see that $s(h(\alpha))$ is increasing as $\epsilon$ increases whenever $0 < \epsilon$, and hence the smoothness $s(h(\alpha))$ is increasing as $\alpha$ decreases whenever $cx_n \leq \alpha < cx_1$.

For the case $j' = l$ where $\delta_l = x_1 - x_n < \epsilon$, we have $x_n - \alpha/c = x_n - (x_1 - \epsilon) = \epsilon - (x_1 - x_n) > 0$ which implies $\alpha < cx_n$ and $h(\alpha) = z(\alpha)$. We have shown that the smoothness is increasing as $\alpha$ is going far from $\langle z, e \rangle$; in particular, when $\alpha < \langle z, e \rangle$ and $\alpha$ is deceasing. One can check that

$$cx_n = \frac{\sum_{i=1}^n d_i x_n}{c} = \left\langle x_n u_n, \frac{\tilde{D} u_n}{c} \right\rangle \leq \left\langle x, \frac{\tilde{D} u_n}{c} \right\rangle = \left\langle \tilde{D}^{\frac{1}{2}} x, \frac{\tilde{D}^{\frac{1}{2}} u_n}{c} \right\rangle = \langle z, e \rangle$$

which means the smoothness is increasing as $\alpha$ decreases whenever $\alpha < cx_n$.

We conclude that the smoothness is increasing as $\alpha$ decreases whenever $\alpha < cx_1$. On the other hand, we have $\sup_{\alpha < cx_1} s(h(\alpha)) = 1$ as the case in the proof of Proposition C.1. One can check that $s(h(\alpha))$ is a continuous function for $\alpha < cx_1$ and thus it has range $[K_1/c, 1)$ by the mean value theorem.

Finally, we can establish the result: $K_1/c = \sqrt{\frac{\sum_{x_i = \max x} d_i}{\sum_{j=1}^n d_j}}$ is the minimum of $s(h(\alpha))$ and 1 is the maximum of $s(h(\alpha))$ occurring whenever $\alpha \geq cx_1 = \sqrt{\sum_{j=1}^n d_j} \max_i x_i$. Moreover, $s(h(\alpha))$ has a monotone property when $\alpha < \sqrt{\sum_{j=1}^n d_j} \max_i x_i$ and has range $\left[ \sqrt{\frac{\sum_{x_i = \max x} d_i}{\sum_{j=1}^n d_j}}, 1 \right]$.

It is clear that the assumption on the ordering of the entries of $x$ will not affect this result. $\qquad \square$

To prove Proposition 4.3, we first prove an analogous result for the identity function, that is, $h = \sigma(z) = z$.

**Proposition C.1.** *Suppose $z_{\mathcal{M}^\perp} \neq 0$, then $s(z(\alpha))$ achieves its minimum $0$ if $\alpha = \langle z, e \rangle$. Moreover, $\sup_\alpha s(z(\alpha)) = 1$ where $s(z(\alpha))$ is close to 1 when $\alpha$ is far away from $\langle z, e \rangle$.*

Notice that Proposition C.1 does not consider the activation function.

*Proof of Proposition C.1.* We know that $0 \leq s(z(\alpha)) \leq 1$ and

$$s(z(\alpha)) = \sqrt{1 - \frac{\|z_{\mathcal{M}^\perp}\|^2}{\|z(\alpha)\|^2}}$$

$$= \sqrt{1 - \frac{\|z_{\mathcal{M}^\perp}\|^2}{\|z_{\mathcal{M}^\perp}\|^2 + \|z(\alpha)_{\mathcal{M}}\|^2}}$$

$$= \sqrt{1 - \frac{\|z_{\mathcal{M}^\perp}\|^2}{\|z_{\mathcal{M}^\perp}\|^2 + \|z_{\mathcal{M}} - \alpha e\|^2}}.$$

Suppose $s(z(\alpha)) = 1$. Then we have

$$\frac{\|z_{\mathcal{M}^\perp}\|^2}{\|z_{\mathcal{M}^\perp}\|^2 + \|z_{\mathcal{M}} - \alpha e\|^2} = 0$$

which forces $\|z_{\mathcal{M}^\perp}\| = 0$. However, this contradicts to the hypothesis $z_{\mathcal{M}^\perp} \neq 0$. So $s(z(\alpha))$ cannot attain its maximum.

But for any $0 \leq t < 1$, one can see that $s(\boldsymbol{z}(\alpha)) = t$ if and only if

$$\sqrt{1 - \frac{\|\boldsymbol{z}_{\mathcal{M}^\perp}\|^2}{\|\boldsymbol{z}_{\mathcal{M}^\perp}\|^2 + \|\boldsymbol{z}_{\mathcal{M}} - \alpha\boldsymbol{e}\|^2}} = t$$

$$\Leftrightarrow \frac{\|\boldsymbol{z}_{\mathcal{M}^\perp}\|^2}{\|\boldsymbol{z}_{\mathcal{M}^\perp}\|^2 + \|\boldsymbol{z}_{\mathcal{M}} - \alpha\boldsymbol{e}\|^2} = 1 - t^2$$

$$\Leftrightarrow \|\boldsymbol{z}_{\mathcal{M}^\perp}\|^2 = (1 - t^2)\big(\|\boldsymbol{z}_{\mathcal{M}^\perp}\|^2 + \|\boldsymbol{z}_{\mathcal{M}} - \alpha\boldsymbol{e}\|^2\big)$$

$$\Leftrightarrow t^2\|\boldsymbol{z}_{\mathcal{M}^\perp}\|^2 = (1 - t^2)\|\boldsymbol{z}_{\mathcal{M}} - \alpha\boldsymbol{e}\|^2$$

$$\Leftrightarrow \|\boldsymbol{z}_{\mathcal{M}} - \alpha\boldsymbol{e}\| = \sqrt{\frac{t^2}{1 - t^2}} \cdot \|\boldsymbol{z}_{\mathcal{M}^\perp}\|$$

This implies that $\sup_\alpha s(\boldsymbol{z}(\alpha)) = 1$ and $s(\boldsymbol{z}(\alpha))$ achieves its minimum $0$ if and only if $\alpha = \langle \boldsymbol{z}, \boldsymbol{e} \rangle$. It is clear that $s(\boldsymbol{z}(\alpha))$ get closer to 1 when $\alpha$ is going far away from $\langle \boldsymbol{z}, \boldsymbol{e} \rangle$. i.e. $|\alpha - \langle \boldsymbol{z}, \boldsymbol{e} \rangle| = \|\boldsymbol{z}_{\mathcal{M}} - \alpha\boldsymbol{e}\|$ is increasing. $\qquad\square$

*Proof of Proposition 4.3.* First, we notice that leaky ReLU has the following two properties:

1. $\sigma_a(x) > 0$ for $x \gg 0$ and $\sigma_a(x) < 0$ for $x \ll 0$.

2. $\sigma_a$ is a non-trivial linear map for $x \gg 0$.

We will use Property 1 to show that $\min_\alpha s(\boldsymbol{h}(\alpha)) = 0$ and Property 2 to show that $\sup_\alpha s(\boldsymbol{h}(\alpha)) = 1$. Notice that $\sigma_a(x) < 0$ for $x \ll 0$ implies that there exists a sufficient small $\alpha_2 < 0$ s.t. all of the entries of $\boldsymbol{h}(\alpha_2)$ are negative and hence $|\langle \boldsymbol{h}(\alpha_2), \boldsymbol{e} \rangle| < 0$. Similarly, $\sigma_a(x) > 0$ for $x \gg 0$ implies that there exists a sufficient large $\alpha_1 > 0$ s.t. all of the entries of $\boldsymbol{h}(\alpha_1)$ are positive and hence $|\langle \boldsymbol{h}(\alpha_1), \boldsymbol{e} \rangle| > 0$. Since $|\langle \boldsymbol{h}(\alpha), \boldsymbol{e} \rangle|$ is a continuous function of $\alpha$ on $[\alpha_1, \alpha_2]$, the Intermediate Value Theorem follows that there exists an $\alpha \in (\alpha_1, \alpha_2)$ s.t. $|\langle \boldsymbol{h}(\alpha), \boldsymbol{e} \rangle| = 0$. Thus by definition $s(\boldsymbol{h}(\alpha)) = |\langle \boldsymbol{h}(\alpha), \boldsymbol{e} \rangle| / \|\boldsymbol{h}(\alpha)\|$, we see that $\min_\alpha s(\boldsymbol{h}(\alpha)) = 0$.

On the other hand, since $\sigma_a$ is a non-trivial linear map for $x \gg 0$, we may assume $\sigma_a(x) = cx$ for $x > x_0$ where $c \neq 0$ is some non-zero constant and $x_0 > 0$ is some positive constant. Then we can choose an $\alpha_0 > \langle \boldsymbol{z}, \boldsymbol{e} \rangle$ s.t. for any $\alpha \geq \alpha_0$, all of the entries of $\boldsymbol{z}(\alpha)$ are greater than $x_0$. Then whenever $\alpha \geq \alpha_0$, we have $\boldsymbol{h}(\alpha) = \sigma_a(\boldsymbol{z}(\alpha)) = c\boldsymbol{z}(\alpha)$. This implies

$$s(\boldsymbol{h}(\alpha)) = \frac{|\langle \boldsymbol{h}(\alpha), \boldsymbol{e} \rangle|}{\|\boldsymbol{h}(\alpha)\|} = \frac{|\langle c\boldsymbol{z}(\alpha), \boldsymbol{e} \rangle|}{\|c\boldsymbol{z}(\alpha)\|} = \frac{|\langle \boldsymbol{z}(\alpha), \boldsymbol{e} \rangle|}{\|\boldsymbol{z}(\alpha)\|} = s(\boldsymbol{z}(\alpha)).$$

Thus $\sup_\alpha s(\boldsymbol{h}(\alpha)) = 1$ follows from the Proof of Proposition C.1 where we see that $\sup_\alpha s(\boldsymbol{z}(\alpha)) = 1$ since $s(\boldsymbol{z}(\alpha))$ gets closer to 1 as $\alpha$ increases.

$\qquad\square$

**Remark C.2.** *Indeed, it holds for any continuous function $f : \mathbb{R} \to \mathbb{R}$ satisfying the following conditions:*

1. *$f(x) > 0$ for $x \gg 0$, $f(x) < 0$ for $x \ll 0$ or $f(x) < 0$ for $x \gg 0$, $f(x) > 0$ for $x \ll 0$,*

2. *$f$ is a non-trivial linear map for $x \gg 0$ or $x \ll 0$.*

*One can check the proof above only depends on these two properties. It is worth mentioning that most activation functions, e.g. leaky LU, SiLU, $\tanh$, satisfy condition 1.*

*Proof of Corollary 4.5.* For any $\alpha$, we notice that $\|\boldsymbol{z}\|_{\mathcal{M}^\perp} = \|\boldsymbol{z}_{\mathcal{M}^\perp}\|_F = \|\boldsymbol{z}(\alpha)\|_{\mathcal{M}^\perp}$ since $\alpha$ only changes the component of $\boldsymbol{z}$ in the eigenspace $\mathcal{M}$. Also, Propositions 3.2 and 3.3 show that $\|\boldsymbol{z}(\alpha)\|_{\mathcal{M}^\perp} \geq \|\boldsymbol{h}(\alpha)\|_{\mathcal{M}^\perp}$ whenever $\boldsymbol{h}(\alpha) = \sigma(\boldsymbol{z}(\alpha))$ or $\sigma_a(\boldsymbol{z}(\alpha))$. Therefore, we see that $\|\boldsymbol{z}\|_{\mathcal{M}^\perp} \geq \|\boldsymbol{h}(\alpha)\|_{\mathcal{M}^\perp}$ holds for any $\alpha$.

Since $\boldsymbol{z}_{\mathcal{M}^\perp} \neq 0$, $s(\boldsymbol{z})$ must lie in $[0, 1)$

$\qquad\square$

## D EXPERIMENTAL DETAILS

This part includes the missing details about the experimental configurations and additional experimental results for Section 6. All tasks we run using Nvidia RTX 3090, GV100, and Tesla T4 GPUs. For consistency, all computational performance metrics, including timing procedures, are run using Tesla T4 GPUs from Google Colab.

### D.1 DATASET DETAILS

In this section, we briefly describe the benchmark datasets used. Table 4 provides additional details about the underlying graph representation.

**Citation Datasets:** The five citation datasets considered are Cora, Citeseer PubMed, Coauthor-Physics, and Ogbn-arxiv. Each dataset is represented by a graph with nodes representing academic publications, features encoding a bag-of-words description, labels classifying the publication type, and edges representing citations.

**Web Knowledge-Base Datasets:** The three web knowledge-base datasets are Cornell, Texas, and Wisconsin. Each dataset is represented by a graph with nodes representing CS department webpages, features encoding a bag-of-words description, edges representing hyper-link connections, and labels classifying the webpage type.

**Wikipedia Network Datasets:** The two Wikipedia network datasets are Chameleon and Squirrel. Each dataset is represented by a graph with nodes representing CS department webpages, features encoding a bag-of-words description, edges representing hyper-link connections, and labels classifying the webpage type.

| | # Nodes | # Edges | # Features | # Classes | Splits (Train/Val/Test) |
|---|---|---|---|---|---|
| Cornell | 183 | 295 | 1,703 | 5 | 48/32/20% |
| Texas | 181 | 309 | 1,703 | 5 | 48/32/20% |
| Wisconsin | 251 | 499 | 1,703 | 5 | 48/32/20% |
| Chameleon | 2,277 | 36,101 | 2,325 | 5 | 48/32/20% |
| Squirrel | 5,201 | 217,073 | 2,089 | 5 | 48/32/20% |
| Citeseer | 3,727 | 4,732 | 3,703 | 6 | 120/500/1000 |
| Cora | 2,708 | 5,429 | 1,433 | 7 | 140/500/1000 |
| PubMed | 19,717 | 44,338 | 500 | 3 | 60/500/1000 |
| Coauthor-Physics | 34,493 | 247,962 | 8415 | 5 | 100/150/34,243 |
| Ogbn-arxiv | 169,343 | 1,166,243 | 128 | 40 | 90,941/29,799/48,603 |
| Roman-Empire | 22,662 | 32,927 | 300 | 18 | 50/25/25% |

Table 4: Graph statistics.

### D.2 MODEL SIZE AND COMPUTATIONAL TIME FOR CITATION DATASETS

Table 5 compares the model size and computational time (both training and testing) for experiments on citation datasets in Section 6.2.

### D.3 ADDITIONAL SECTION 6.2 DETAILS FOR CITATION DATASETS

Table 6 lists the hyperparameters used in the grid search for each model in generating the results in Table 1. Table 10 reports the classification accuracy of different models with different depths using either ReLU or leaky ReLU activation function.

To test the significance of the SCT based models, we perform a trial of 100 random model initializations. Table8 reports the mean test accuracy $\pm$ standard deviation for each model of varying depth. We compare against each of the baseline models generating results for GCN and EGNN, and reporting results from [8] for GCNII (marked with $^*$). We then perform a t-test at $0.95$ confidence, where

$$\text{t-score} = \frac{\mu_{*\text{-SCT}} - \mu_*}{\sqrt{\frac{\sigma_{*\text{-SCT}}^2}{n} + \frac{\sigma_*^2}{n}}}$$

| | # Parameters | Training Time (s) | Inference Time (ms) |
|---|---|---|---|
| **Cora** | | | |
| GCN | 100,423 | 8.4 | 1.6 |
| GCNII | 110,535 | 10.0 | 2.1 |
| GCNII | 708,743 | 57.6 | 12.3 |
| GCNII-SCT | 1,237,127 | 110.3 | 29.6 |
| EGNN | 712,839 | 65.6 | 14.4 |
| EGNN-SCT | 316,551 | 24.8 | 4.5 |
| **Citeseer** | | | |
| GCN | 245,638 | 8.3 | 1.5 |
| GCN-SCT | 301,830 | 15.5 | 4.0 |
| GCNII | 999,174 | 57.6 | 12.3 |
| GCNII-SCT | 1,001,222 | 65.9 | 15.7 |
| EGNN | 739,078 | 39.6 | 7.2 |
| EGNN-SCT | 540,934 | 24.0 | 5.8 |
| **PubMed** | | | |
| GCN | 40,451 | 9.0 | 1.8 |
| GCN-SCT | 40,707 | 11.1 | 2.2 |
| GCNII | 326,659 | 98.2 | 12.8 |
| GCNII-SCT | 590,851 | 71.7 | 17.4 |
| EGNN | 592,899 | 93.7 | 2.5 |
| EGNN-SCT | 130,563 | 16.0 | 3.1 |
| **Coauthor-Physics** | | | |
| GCN | 547,141 | 35.2 | 8.0 |
| GCN-SCT | 547,397 | 33.9 | 8.3 |
| GCNII | 555,333 | 49.1 | 10.3 |
| GCNII-SCT | 555,461 | 67.0 | 9.5 |
| EGNN | 672,069 | 176.4 | 47.9 |
| EGNN-SCT | 572,229 | 51.7 | 14.8 |
| **Ogbn-arxiv** | | | |
| GCN | 27,240 | 50.4 | 21.1 |
| GCN-SCT | 28,392 | 62.6 | 24.4 |
| GCNII | 76,392 | 205.4 | 94.8 |
| GCNII-SCT | 80,616 | 253.0 | 108.9 |
| EGNN | 77,416 | 206.8 | 98.0 |
| EGNN-SCT | 81,640 | 254.0 | 112.3 |

Table 5: Number of model parameters for varying numbers of layers using the optimal model hyperparameters. The SCT is added at each layer and the size of the additional parameters scales with the number of eigenvectors with an eigenvalue of one for matrix $G$ in equation (2).

| Parameter | Values |
|---|---|
| Learning Rate | $\{1e\text{-}4, 1e\text{-}3, 1e\text{-}2\}$ |
| Weight Decay (FC) | $\{0, 1e\text{-}4, 5e\text{-}4, 1e\text{-}3, 5e\text{-}3, 1e\text{-}2\}$ |
| Weight Decay (Conv) | $\{0, 1e\text{-}4, 5e\text{-}4, 1e\text{-}3, 5e\text{-}3, 1e\text{-}2\}$ |
| Dropout | $\{0.1, 0.2, 0.3, 0.4, 0.5, 0.6, 0.7, 0.8, 0.9\}$ |
| Hidden Channels | $\{16, 32, 64, 128\}$ |
| GCNII-$\alpha$ | $\{0.1, 0.2, 0.3, 0.4, 0.5, 0.6, 0.7, 0.8, 0.9\}$ |
| GCNII-$\theta$ | $\{0.1, 0.2, 0.3, 0.4, 0.5, 0.6, 0.7, 0.8, 0.9\}$ |
| EGNN-$c_{max}$ | $\{0.5, 1.0, 1.5, 2.0\}$ |
| EGNN-$\alpha$ | $\{0.1, 0.2, 0.3, 0.4, 0.5, 0.6, 0.7, 0.8, 0.9\}$ |
| EGNN-$\theta$ | $\{0.1, 0.2, 0.3, 0.4, 0.5, 0.6, 0.7, 0.8, 0.9\}$ |

Table 6: Hyperparameter grid search for Table 1.

The results of the t-test are reported in Table 9.

| Layers | 2 | 4 | 16 | 32 |
|---|---|---|---|---|
| **Cora** | | | | |
| EGNN/EGNN-SCT | 83.2/**83.4** | 84.2/**84.3** | 85.4/**85.5** | 85.3/**85.5** |
| **Citeseer** | | | | |
| EGNN/EGNN-SCT | 72.0/**72.1** | 71.9/**72.3** | 72.4/**72.6** | 72.3/**72.8** |
| **PubMed** | | | | |
| EGNN/EGNN-SCT | 79.2/**79.4** | 79.5/**79.8** | **80.1**/80.1 | 80.0/**80.2** |
| **Coauthor-Physics** | | | | |
| EGNN/EGNN-SCT | 92.6/**92.8** | 92.9/**93.0** | 93.1/**93.3** | **93.3**/93.3 |
| **Ogbn-arxiv** | | | | |
| EGNN/EGNN-SCT | 68.4/**68.5** | 71.1/**71.3** | 72.7/**73.0** | 72.7/**72.9** |

Table 7: Test accuracy for EGNN and EGNN-SCT using SReLU activation function of varying depth on citation networks with the split discussed in Section 6.2. (Unit:%)

| Layers | 2 | 4 | 16 | 32 |
|---|---|---|---|---|
| **Cora** | | | | |
| GCN | $78.45 \pm 2.29$ | $71.26 \pm 8.09$ | $55.82 \pm 5.10$ | $30.52 \pm 4.48$ |
| GCN-SCT | $\mathbf{81.88 \pm 0.98}$ | $\mathbf{78.54 \pm 2.54}$ | $\mathbf{60.51 \pm 5.26}$ | $\mathbf{40.30 \pm 15.81}$ |
| GCNII* | $82.19 \pm 0.77$ | $82.84 \pm 0.61$ | $84.69 \pm 0.51$ | $85.29 \pm 0.47$ |
| GCNII-SCT | $\mathbf{83.67 \pm 0.45}$ | $\mathbf{83.27 \pm 0.41}$ | $\mathbf{84.73 \pm 0.63}$ | $\mathbf{85.32 \pm 0.64}$ |
| EGNN | $83.16 \pm 0.38$ | $84.17 \pm 0.36$ | $85.36 \pm 0.45$ | $\mathbf{85.43 \pm 0.41}$ |
| EGNN-SCT | $\mathbf{83.56 \pm 0.40}$ | $\mathbf{84.35 \pm 0.47}$ | $\mathbf{85.37 \pm 0.52}$ | $85.36 \pm 0.46$ |
| **Citeseer** | | | | |
| GCN | $65.33 \pm 1.74$ | $56.57 \pm 4.21$ | $18.24 \pm 1.72$ | $29.67 \pm 6.46$ |
| GCN-SCT | $\mathbf{65.47 \pm 1.70}$ | $\mathbf{64.86 \pm 1.58}$ | $\mathbf{50.89 \pm 3.49}$ | $\mathbf{43.07 \pm 4.68}$ |
| GCNII* | $67.81 \pm 0.89$ | $68.10 \pm 0.84$ | $\mathbf{72.97 \pm 0.71}$ | $73.24 \pm 0.78$ |
| GCNII-SCT | $\mathbf{71.25 \pm 0.96}$ | $\mathbf{69.66 \pm 1.78}$ | $72.86 \pm 0.74$ | $\mathbf{73.30 \pm 1.33}$ |
| EGNN | $71.82 \pm 0.49$ | $72.04 \pm 0.50$ | $72.52 \pm 0.67$ | $72.54 \pm 0.65$ |
| EGNN-SCT | $\mathbf{72.88 \pm 0.50}$ | $\mathbf{72.05 \pm 0.58}$ | $\mathbf{72.57 \pm 0.87}$ | $\mathbf{72.69 \pm 0.72}$ |
| **PubMed** | | | | |
| GCN | $\mathbf{77.43 \pm 0.90}$ | $75.63 \pm 3.72$ | $40.85 \pm 5.09$ | $41.11 \pm 1.77$ |
| GCN-SCT | $77.32 \pm 1.20$ | $\mathbf{76.46 \pm 2.53}$ | $\mathbf{64.85 \pm 13.14}$ | $\mathbf{66.27 \pm 10.80}$ |
| GCNII* | $78.05 \pm 1.53$ | $77.86 \pm 0.91$ | $80.03 \pm 0.50$ | $79.91 \pm 0.27$ |
| GCNII-SCT | $\mathbf{78.36 \pm 0.59}$ | $\mathbf{77.89 \pm 1.08}$ | $\mathbf{80.65 \pm 0.41}$ | $\mathbf{80.51 \pm 0.64}$ |
| EGNN | $79.27 \pm 0.37$ | $79.51 \pm 0.30$ | $79.88 \pm 0.27$ | $79.92 \pm 0.28$ |
| EGNN-SCT | $\mathbf{79.35 \pm 0.37}$ | $\mathbf{79.70 \pm 0.33}$ | $\mathbf{80.07 \pm 0.35}$ | $\mathbf{80.03 \pm 0.30}$ |

Table 8: Mean test accuracy $\pm$ standard deviation over 100 random initializations for each model of varying depth. We compare against each of the baseline models generating results for GCN and EGNN, and report the results (*) for GCNII from [8]. SCT improves almost all baseline models except 32-layer EGNN for Cora, 16-layer GCNII for Citeseer, and 2-layer GCN for PubMed. In these three cases, models with SCT are only marginally inferior to the baseline models.

| Layers | 2 | 4 | 16 | 32 |
|---|---|---|---|---|
| **Cora** | | | | |
| GCN-SCT | 13.77 | 8.72 | 6.40 | 5.95 |
| GCNII-SCT | 16.59 | 5.85 | 0.49 | 0.38 |
| EGNN-SCT | 7.29 | 3.04 | 0.15 | $-1.16$ |
| **Citeseer** | | | | |
| GCN-SCT | 0.58 | 18.44 | 83.92 | 16.58 |
| GCNII-SCT | 26.28 | 7.93 | $-1.07$ | 0.39 |
| EGNN-SCT | 15.14 | 0.13 | 0.46 | 1.55 |
| **PubMed** | | | | |
| GCN-SCT | $-0.77$ | 1.84 | 17.03 | 22.99 |
| GCNII-SCT | 1.89 | 0.21 | 9.51 | 8.64 |
| EGNN-SCT | 1.57 | 4.26 | 4.30 | 2.68 |

Table 9: We conduct t-test experiments at 0.95 confidence to compare models with and without SCT on different benchmark graph node classification tasks. We observe that in general SCT provides significant improvements, and only fails to improve in very few cases and by a marginal amount. A larger t-score means a more significant improvement.

### D.3.1 VANISHING GRADIENTS

Figure 4 shows the vanishing gradient problem for training deep GCN – with or without SCT – in comparison to models like GCNII and EGNN. This figure plots $||\partial \boldsymbol{H}^{\text{out}}/\partial \boldsymbol{H}^l||$ for layers $l \in [0, 32]$ as the training epochs run from 0 to 100. Figures 4 (a) and (b) illustrate the vanishing gradient issue for GCN and that it persists for GCN-SCT. Figures 4 (c) and (e) illustrate that GCNII and EGNN do not suffer from vanishing gradients, and furthermore, because these models connect $\boldsymbol{H}^0$ to every layer, the gradient with respect to the weights in the first layer is nonzero. What is interesting about the addition of SCT to both EGNN and GCNII is that the intermediate gradients become large as the training epochs progress shown in Figure 4 (d) and (f).

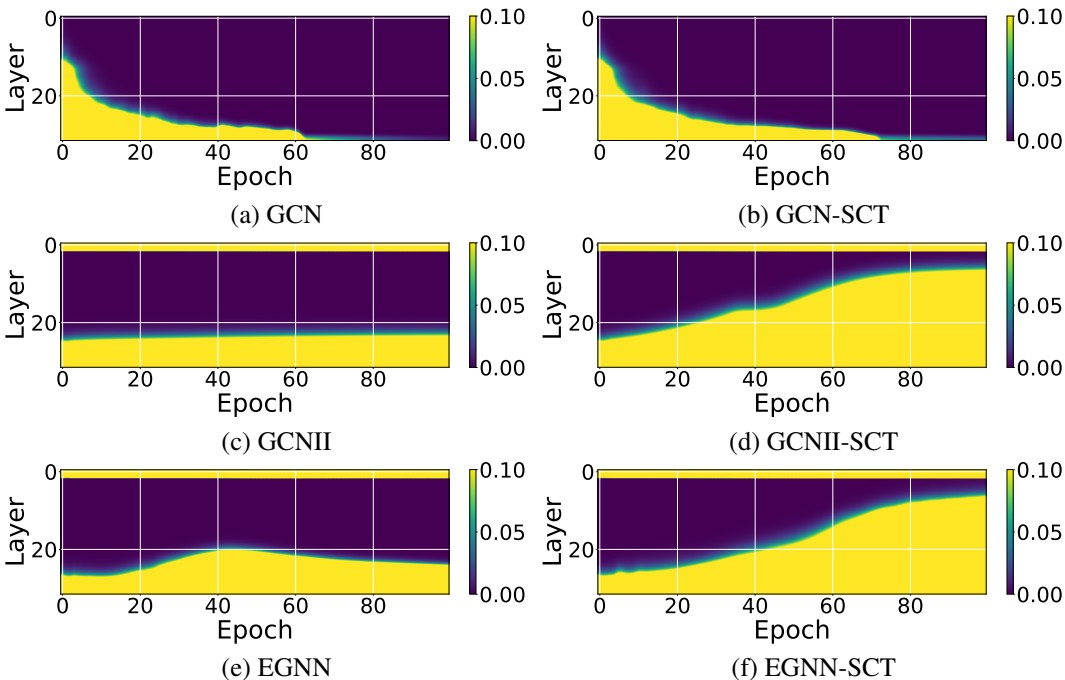

Figure 4: Training gradients for $||\partial \boldsymbol{H}^{\text{out}}/\partial \boldsymbol{H}^l||$ for $l \in [0, 32]$ layers and 100 training epochs on the Citeseer dataset. Here, all models have 32 layers and 16 hidden dimensions for each layer. We observe that (a) GCN suffers from vanishing gradients. By contrast (c) GCNII and (e) EGNN do not suffer from vanishing gradients, and we can observe their skip connection to $\boldsymbol{H}^0$. Because these models (GCNII/GCNII-SCT and EGNN/EGNN-SCT) connect $\boldsymbol{H}^0$ to every layer, the gradient at the first layer is nonzero. We notice that while SCT does not overcome vanishing gradients for (b) GCN-SCT, it is able to increase the norm of the gradients for the intermediate layers in (d) GCNII-SCT and (f) EGNN-SCT.

### D.4 ADDITIONAL SECTION 6.2 DETAILS FOR OTHER DATASETS

Table 11 reports the mean and standard deviation test accuracy over ten folds of the WebKB and WikipediaNetwork datasets using the SCT-based models.

Table 12 lists the average computational time for each epoch for different models of the same depth – 8 layers. These results show that integrating SCT into GNNs only results in a small amount of computational overhead.

### D.5 ADDITIONAL RESULTS FOR HETEROPHILIC GRAPHS

Heterophilic graphs contain a higher connection likelihood between distinct rather than similar labels. Following the experimental setup described in [30], we compare GCN/GCN-SCT and GCNII/GCNII-SCT using 10 fixed splits of 50/25/25% for training, validation, and testing, respectively. As in [30], we report the mean accuracy and standard deviation for 10-fold cross validation on the test results in Table 13. We utilize the same architectures and hyperparameter tuning as in Section 6.2.

| Cora | | | | | | | | |
|---|---|---|---|---|---|---|---|---|
| | ReLU | | | | leaky ReLU | | | |
| Layers | 2 | 4 | 16 | 32 | 2 | 4 | 16 | 32 |
| GCN-SCT | 81.2 | 80.3 | 71.4 | 67.2 | 82.9 | 82.8 | 68.0 | 65.5 |
| GCNII-SCT | 83.5 | 83.8 | 82.7 | 83.3 | 83.8 | 84.8 | 84.8 | 85.5 |
| EGNN-SCT | 84.1 | 83.8 | 82.3 | 80.8 | 83.7 | 84.5 | 83.3 | 82.0 |

| Citeseer | | | | | | | | |
|---|---|---|---|---|---|---|---|---|
| | ReLU | | | | leaky ReLU | | | |
| Layers | 2 | 4 | 16 | 32 | 2 | 4 | 16 | 32 |
| GCN-SCT | 69.0 | 67.3 | 51.5 | 50.3 | 69.9 | 67.7 | 55.4 | 51.0 |
| GCNII-SCT | 72.8 | 72.8 | 72.8 | 73.3 | 72.8 | 72.9 | 73.8 | 72.7 |
| EGNN-SCT | 72.5 | 72.0 | 70.2 | 71.8 | 73.1 | 71.7 | 72.6 | 72.9 |

| PubMed | | | | | | | | |
|---|---|---|---|---|---|---|---|---|
| | ReLU | | | | leaky ReLU | | | |
| Layers | 2 | 4 | 16 | 32 | 2 | 4 | 16 | 32 |
| GCN-SCT | 79.4 | 78.2 | 75.9 | 77.0 | 79.8 | 78.4 | 76.1 | 76.9 |
| GCNII-SCT | 79.7 | 80.1 | 80.7 | 80.7 | 79.6 | 80.0 | 80.3 | 80.7 |
| EGNN-SCT | 79.7 | 80.1 | 80.0 | 80.4 | 79.8 | 80.4 | 80.3 | 80.2 |

| Coauthor-Physics | | | | | | | | |
|---|---|---|---|---|---|---|---|---|
| | ReLU | | | | leaky ReLU | | | |
| Layers | 2 | 4 | 16 | 32 | 2 | 4 | 16 | 32 |
| GCN-SCT | $91.8 \pm 1.6$ | $91.6 \pm 3.0$ | $44.5 \pm 13.0$ | $42.6 \pm 17.0$ | $92.6 \pm 1.6$ | $92.5 \pm 5.9$ | $50.9 \pm 15.0$ | $43.6 \pm 16.0$ |
| GCNII-SCT | $94.4 \pm 0.4$ | $93.5 \pm 1.2$ | $93.7 \pm 0.7$ | $93.8 \pm 0.6$ | $94.0 \pm 0.4$ | $94.2 \pm 0.3$ | $93.3 \pm 0.7$ | $94.1 \pm 0.3$ |
| EGNN-SCT | $93.6 \pm 0.7$ | $94.1 \pm 0.4$ | $93.4 \pm 0.8$ | $93.8 \pm 1.3$ | $93.9 \pm 0.7$ | $94.0 \pm 0.7$ | $94.0 \pm 0.7$ | $93.3 \pm 0.9$ |

| Ogbn-arxiv | | | | | | | | |
|---|---|---|---|---|---|---|---|---|
| | ReLU | | | | leaky ReLU | | | |
| Layers | 2 | 4 | 16 | 32 | 2 | 4 | 16 | 32 |
| GCN-SCT | $71.7 \pm 0.3$ | $72.6 \pm 0.3$ | $71.4 \pm 0.2$ | $71.9 \pm 0.3$ | $72.1 \pm 0.3$ | $72.7 \pm 0.3$ | $72.3 \pm 0.2$ | $72.3 \pm 0.3$ |
| GCNII-SCT | $71.4 \pm 0.3$ | $72.1 \pm 0.3$ | $72.2 \pm 0.2$ | $71.8 \pm 0.2$ | $72.0 \pm 0.3$ | $72.2 \pm 0.2$ | $72.4 \pm 0.3$ | $72.1 \pm 0.3$ |
| EGNN-SCT | $68.5 \pm 0.6$ | $71.0 \pm 0.5$ | $72.8 \pm 0.5$ | $72.1 \pm 0.6$ | $67.7 \pm 0.5$ | $71.3 \pm 0.5$ | $72.3 \pm 0.5$ | $72.3 \pm 0.5$ |

Table 10: Test accuracy results for models of varying depth with ReLU or leaky ReLU activation function on the citation network datasets using the split discussed in Section 6.2.

| | Cornell | Texas | Wisconsin | Chameleon | Squirrel |
|---|---|---|---|---|---|
| GCN-SCT | $55.95 \pm 8.5$ | $62.16 \pm 5.7$ | $54.71 \pm 4.4$ | $38.44 \pm 4.3$ | $35.31 \pm 1.9$ |
| GCNII-SCT | $75.41 \pm 2.2$ | $83.34 \pm 4.5$ | $86.08 \pm 3.8$ | $64.52 \pm 2.2$ | $47.51 \pm 1.4$ |

Table 11: Test mean $\pm$ standard deviation accuracy from 10 fold cross validation on five heterophilic datasets with fixed $48/32/20\%$ splits. The depth of each model is 8 layers with 16 hidden channels.

| | Cornell | Texas | Wisconsin | Chameleon | Squirrel |
|---|---|---|---|---|---|
| GCN [22] | 0.011 | 0.013 | 0.012 | 0.011 | 0.022 |
| GCNII [7] | 0.017 | 0.018 | 0.017 | 0.013 | 0.022 |
| GCN-SCT | 0.015 | 0.017 | 0.015 | 0.011 | 0.023 |
| GCNII-SCT | 0.017 | 0.018 | 0.017 | 0.020 | 0.025 |

Table 12: Average computational time per epoch for five heterophilic datasets with fixed $48/32/20\%$ splits. The depth of each model is 8 layers with 16 hidden channels. (Unit: second)

| Layers | 2 | 4 | 16 | 32 |
|---|---|---|---|---|
| **Roman-Empire** | | | | |
| GCN [22] | $84.48 \pm 0.53$ | $84.00 \pm 0.71$ | $74.56 \pm 0.75$ | $14.32 \pm 1.02$ |
| GCN-SCT | $\mathbf{85.37 \pm 0.56}$ | $\mathbf{84.08 \pm 0.71}$ | $\mathbf{82.58 \pm 0.57}$ | $\mathbf{79.6 \pm 0.49}$ |
| GCNII[7] | $83.49 \pm 0.36$ | $83.43 \pm 0.40$ | $80.01 \pm 0.50$ | $76.52 \pm 0.70$ |
| GCNII-SCT | $\mathbf{85.44 \pm 0.56}$ | $\mathbf{85.08 \pm 0.24}$ | $\mathbf{81.44 \pm 0.35}$ | $\mathbf{77.28 \pm 0.55}$ |

Table 13: Test mean $\pm$ standard deviation accuracy from 10 fold cross validation on heterophilic Roman-Empire graph with fixed $50/25/25\%$ splits. (Unit:%)

