# OpenReview forum: "Rethinking the Smoothness of Node Features Learned by Graph Convolutional Networks"
_ICLR.cc/2024/Conference — Submitted to ICLR 2024_

### Official Review · Reviewer_pkn8 · 2023-10-30

**Soundness:** 2 fair
**Presentation:** 2 fair
**Contribution:** 2 fair
**Rating:** 3
**Confidence:** 5

**Summary:**

This paper studies the impact of ReLU and LeakyReLU nonlinearities on the smoothness of node features and proposes a method for regulating feature smoothness under a normalized smoothness metric. While earlier theories suggested that these nonlinearities always result in smoother features, this study reveals that, when considering feature magnitude and applying a normalized smoothness metric, ReLU and LeakyReLU can actually increase, decrease, or maintain the smoothness metric. Notably, by adjusting the input's projection in certain eigenspaces, one can manipulate the output's smoothness to achieve a desired level. The paper introduces a technique known as the "smoothness control term" (SCT), which is designed to regulate node feature smoothness, and it is experimentally tested to validate its effectiveness.

**Strengths:**

1. Understanding the effect of nonlinearities is an important question.
2. The empirical performance of SCT looks promising in Table 2.

**Weaknesses:**

1. Although seemingly making sense, “normalized smoothness”, as measured by s defined in eq.(4) is not valid for interpretation. It disconnects node smoothness from model performance, making it carry no insights into practice and thus it is meaningless to study.

    For instance, consider a graph with two classes of nodes: one class having feature values of 1, and the other class with feature values of -1. In this case, a linear classifier would have perfect classification performance.  If we consistently add one to each node's feature, the differences among node features would not change, and if we apply a classifier again to classify based on the new features, the performance would not change either---We are basically shifting all the node features by the same value and the bias term of a classifier can easily accommodate that. Such a phenomenon is well justified by the unnormalized metrics such as conventional Dirichlet energy because it would remain the same before and after we add the same value to each node. However, the normalized smoothness metric s proposed in this paper would get larger and larger, indicating that the node features are getting "smoother" and "smoother".

    Given the above concern and the established research on the effects of ReLU and LeakyReLU under unnormalized smoothness [5, 27] (citations provided by the paper), this paper provides very little new theoretical insight.




2. I also checked [5] (citation provided by the paper), and I didn’t see any serious evidence, either theoretical or empirical, supporting the following highlighted claim in this paper:

> [5] points out that over-smoothing – measured by the distance of node features to the eigenspace M or the Dirichlet energy – is a misnomer, and the real smoothness of a graph signal should be characterized by a normalized smoothness, e.g., normalizing the Dirichlet energy by the magnitude of the features.

The only related sentence I saw was

> Finally, analyzing the real over-smoothing effect, i.e., the Rayleigh quotient $\frac{tr(X^T \tilde{\Delta} X)}{||X||^2_2}$ for
deep GNNs is still an open and important question.

 But this itself doesn't justify the validity of the normalized smoothness.

3.  The improvement over stronger baselines (GCNII and EGNN) in Table 1 is limited in most cases, which raises doubts about the overall effectiveness of SCT.

**Questions:**

Could the authors provide standard deviations for the experimental results in Table 1 (particularly for the baseline methods) and Table 2?

---

> ### Author Response · Authors · 2023-11-15
> **Response to Reviewer pkn8 (1/3)**
>
> We thank the reviewer for the review. However, **we strongly disagree with your highly unethical and insulting comments, which falsify the truth**. We want to stress that respecting truth is the ethics of being a reviewer. Criticizing another's paper by falsifying the truth is scientific misconduct. It is a severe offense that can damage the reputation of other scientists and undermine public trust in science.
>
>  Before addressing your comments, we would like to point out that:
>
> - Your criticism that “studying the normalized smoothness of GCN features is meaningless” is rootless, unethical, and insulting.
> In particular, studying an existing smoothness notion and an existing important open problem is not a weakness.
>
> - ***Evidence for studying normalized smoothness:*** The correlation between the relative magnitude $||{\bf z}\_{\mathcal{M}^\perp}||/||{\bf z}\_\mathcal{M}||$ -- a quantity that is closely related to the normalized smoothness -- and classification accuracy was first empirically studied in Figure 4 by Oono & Suzuki (ICLR 2020). Moreover, Cai & Wang (arXiv:2006.13318) pointed out that studying normalized smoothness is an open and important problem. Again, ***we believe that studying an open and important problem, pointed out in the pioneering paper, as part of our paper is not a weakness.***
>
> ---
>
> Below, we address your comments in detail.
>
> ---
>
> **Studying the normalized smoothness is a weakness**
>
> **Reply:** ***We respectfully but strongly disagree that studying an existing smoothness notion and an existing open problem as part of our work is a weakness.***
>
> The importance of studying normalized smoothness is supported by evidence. ***Empirical evidence:*** The correlation between the relative magnitude $||{\bf z}\_{\mathcal{M}^\perp}||/||{\bf z}\_\mathcal{M}||$ -- a quantity that is closely related to the normalized smoothness -- and classification accuracy was first empirically studied in Figure 4 by Oono & Suzuki (ICLR 2020). While they demonstrate a strong correlation between classification accuracy and $||{\bf z}\_{\mathcal{M}^\perp}||/||{\bf z}\_{\mathcal{M}}||$, their results neither refute nor conclusively establish that performance improves with an increasing $||{\bf z}\_{\mathcal{M}^\perp}||/||{\bf z}\_\mathcal{M}||$. Despite this, we share the common belief that the ratio $||{\bf z}\_{\mathcal{M}^\perp}||/||{\bf z}\_\mathcal{M}||$ or the normalized smoothness $||{\bf z}\_\mathcal{M}||/||{\bf z}||$ correlates with performance. In contrast to intentionally making features unsmooth, our proposed smoothness control term aims to enable the model to control the smoothness of node features automatically. ***Evidence:*** As the reviewer mentioned, the paper [5] states that **“Finally, analyzing the real over-smoothing effect, i.e., the Rayleigh quotient $\frac{tr(X^T\tilde{\Delta}X)}{|\|X||_2^2}$ for deep GNNs is still an open and important question.”** The Rayleigh quotient is the normalized smoothness.
>
> In the following three paragraphs, we would like to clarify three substantial mistakes in your review:
>
> First, ***In the footnote of the first page of the paper [5]***, Cai & Wang state that “Strictly speaking, over-smoothing is a misnomer. As we will show, what is decreasing is $tr(X^T\tilde{\Delta}X)$, not the real smoothness $\frac{tr(X^T\tilde{\Delta}X)}{|\|X||_2^2}$ of graph signal $X$.” Since the distance of node features to the eigenspace $\mathcal{M}$ and the Dirichlet energy are two equivalent seminorms, we rephrase that “[5] points out that over-smoothing -- measured by the distance of node features to the eigenspace $\mathcal{M}$ or the Dirichlet energy -- is a misnomer, and the real smoothness of a graph signal should be characterized by a normalized smoothness, e.g., normalizing the Dirichlet energy by the magnitude of the features.”

---

> ### Author Response · Authors · 2023-11-15
> **Response to Reviewer pkn8 (2/3)**
>
> Second, ***the normalized smoothness is not a notion we proposed, which is one of the two existing smoothness notions for characterizing the smoothness of GCN features*** We disagree with the statement that “normalized smoothness disconnects node smoothness from model performance, making it carry no insights into practice and thus it is meaningless to study.” On the one hand, ***your statement that normalized smoothness is meaningless to study is a falsification*** and counters to the empirical study in Oono & Suzuki (ICLR 2020) and the paper [5]’s statement that it is an open and important problem. On the other hand, ***the example you provided is about linear classifier (we will comment on it later), which is irrelevant to GCN.*** The GCN layer is given by $H^l=\sigma(W^lH^{l-1}G)$ and the last layer’s feature $H^L$ is directly passed to log\_softmax to get the prediction softmax score and there is no linear layer; see https://github.com/tkipf/pygcn/blob/master/pygcn/models.py. Notice that there is no bias term in the GCN layer. Even after adding a bias term to make a new layer $H^l=\sigma(W^lH^{l-1}G+B)$, how can you learn a bias term, start from random initialization, to easily accommodate the shift in the features -- say $H^{l-1}$ -- while taking into account the multiplication by $G$?
>
> Third, we disagree with your comment that the paper provides very little new theoretical insight. ***Our study both unnormalized and normalized smoothness.*** The paper [5,27] shows that over-smoothing, measured by unnormalized smoothness, is inevitable for GCNs. The results in [5,27] do not imply an approach to control the smoothness of node features. Compared to [5,27], our paper provides important new insights ***both theoretically and practically***: 1) In Section 3, we establish ***a geometric characterization*** of how ReLU and Leaky ReLU affect the smoothness of node features, showing that ***the smoothness of node features are controllable*** by adjusting the projection of node features onto the eigenspace $\mathcal{M}$, which informs the practical smoothness control term. 2) In Section 4, we show that adjusting the projection of node features onto the eigenspace $\mathcal{M}$ results in ***disparate effects on the normalized and unnormalized smoothness***. 3) We study an open problem proposed in [5], which was considered to be important by the authors of [5].
>
>
> ---
>
> **Normalized smoothness lacks interpretability of the classification results of the linear classifier.**
>
> **Reply:** Though irrelevant to our work, we would like to comment on your comment that normalized smoothness lacks interpretability of the classification results of the linear classifier, say $softmax(Wx+b)$. If the features of the training set are shifted, the bias $b$ can accommodate it during training. However, notice that the linear classifier is first trained and then applied to new data. Once the model is trained, we will have a fixed set of weights $W$ and $b$. For fixed $W$ and $b$, $softmax(Wx+b)$ is changed when $x$ is shifted. ***We are not changing $b$ for every new $x$.***
>
> ---
>
> **The improvement over stronger baselines (GCNII and EGNN) in Table 1 is limited in most cases. Could the authors provide standard deviations for the experimental results in Table 1 (particularly for the baseline methods) and Table 2?**
>
> **Reply:**
> ***Standard deviation:*** In our submission, the standard deviations of the models with SCT have been reported when they exist under the benchmark experimental setting. The standard deviation only exists for the experiments of Coauthor-Physics and Ogbn-arxiv in Table 1 and exists for all experiments in Table 2. We stress the benchmark experimental setting here again that we use the fixed public split for the Citation datasets and the particular splits from reference [37] in the revision in the Coauthor-Physics and Ogbn-arxiv datasets. For the baseline models without SCT, the standard deviation results are not reported in the benchmark papers [7,42] in the revision. The code released by the authors of the benchmark papers does not include the hyperparameters used, and we cannot get the exact accuracies by running their released code. For appropriate comparisons, we use the accuracy reported in the benchmark papers.

---

> ### Author Response · Authors · 2023-11-15
> **Response to Reviewer pkn8 (3/3)**
>
> ***Improvement over GCNII and EGNN:*** GCNII is a state-of-the-art GCN-style model and EGNN can be considered as GCNII with constraints on the Dirichlet energy of the node features. The proposed smoothness control term (SCT) can consistently improve the state-of-the-art baseline models, especially on heterophilic graphs. For homophilic graphs, the improvement is more significant on larger-scale datasets -- Coauthor-Physics and Ogbn-arxiv -- than on the smaller-scale datasets -- Cora, Citeseer, and PubMed. In particular, Table 1 in our paper shows that SCT can often improve the accuracy of GCNII and EGNN by more than 1% on Coauthor-Physics and Ogbn-arxiv. As far as we are aware, the performance of GCNII-SCT and EGNN-SCT is the state-of-the-art accuracy for GCN-style models; see Section 5.1 and Appendix A.2 in reference [1] listed below for details.
>
> In Table 1, though the accuracy gain is not huge in absolute values for GCNII and EGNN in some tasks, they are significant in almost all cases. To show the statistical significance of the improvement using SCT for smaller-scale datasets, we have included statistical significance tests for accuracy improvement for Cora, PubMed, and Citeseer. In particular, we train each model with 100 different random initializations using the optimal hyperparameters; the mean and standard deviation of the test accuracy are provided in Table 8 of the revision. Moreover, the t-test results on the statistical significance of the accuracy improvement are provided in Table 9 of the revision.
>
> [1] Zhou et al. Dirichlet Energy Constrained Learning for Deep Graph Neural Network, NeurIPS 2021.
>
>
> ---
>
> We have updated our submission based on the reviewer's feedback, with the revision highlighted in blue.

---

### Official Review · Reviewer_h1DY · 2023-10-31

**Soundness:** 3 good
**Presentation:** 2 fair
**Contribution:** 2 fair
**Rating:** 5
**Confidence:** 3

**Summary:**

This paper investigates the properties of node features learned by Graph Convolutional Networks (GCNs) with a focus on the smoothness of the features. It challenges the conventional understanding of the role of activation functions like ReLU or leaky ReLU in the smoothing process of node features in GCNs. Traditionally, it was believed that these activation functions contribute to smoothing the node features, which is beneficial for tasks like node classification when using a limited number of Graph Convolutional Layers (GCLs). However, the authors argue that this might not always be the case, especially in deeper GCNs. Through empirical studies and theoretical analysis, the paper presents evidence that in deeper networks, the node features might actually become less smooth, contrary to the established belief. This finding is significant as it opens up new avenues for understanding and improving the learning process in deep GCNs, particularly concerning the choice and role of activation functions in shaping the learned node features.

**Strengths:**

* The argument that challenges the traditional understanding of activation function roles in GCNs seems to be presented clearly and logically, enhancing the paper’s accessibility and impact.

* The paper appears to delve deeply into the nuances of node feature smoothness in GCNs, providing a comprehensive analysis that bolsters the quality of the work.

**Weaknesses:**

* The paper seems to heavily rely on previous works [1,2] for its theoretical results. A more independent theoretical contribution or a clearer delineation of the novel aspects beyond the referenced works would strengthen the paper's originality.

* The empirical validation could be broadened to enhance the robustness of the findings. Incorporating a more diverse array of datasets and experimenting with various network architectures would be beneficial. Notably, the largest non-homophily graph used is the Squirrel dataset, which consists of 5201 nodes. Exploring larger and more varied graphs could provide more comprehensive insights.

* The proposed method, as represented by Equation 6, appears to be a somewhat incremental modification, seemingly adding only a bias term to the graph layer. A deeper discussion on the novelty and impact of this modification would be beneficial to understand its significance and contribution better.

* The presentation of results in Table 1 could be improved for clarity and comprehensiveness. Enhancing the table's presentation could make the findings more accessible and effectively communicate the research outcomes.

* The benifit of proposed method is somewhat weak when nerual network is deep.


[1] Chen Cai and Yusu Wang. A note on over-smoothing for graph neural networks. arXiv preprint arXiv:2006.13318, 2020.

[2] Kenta Oono and Taiji Suzuki. Graph neural networks exponentially lose expressive power for node classification. In International Conference on Learning Representations, 2020.

**Questions:**

* Could you clarify the specific novel contributions of your theoretical analysis beyond the foundations laid by references [1,2]?

* Have you considered testing your approach on a broader variety of datasets, especially larger and more complex non-homophily graphs beyond the Squirrel dataset?

* Could you elaborate on the novelty and significance of the modification introduced in Equation 6? How does the addition of a bias term fundamentally impact the model's behavior or performance?

* It seems that the benefits of the proposed method diminish in deeper neural networks. Could you provide more insights into why this might be the case and whether there are ways to mitigate this limitation?


-----------------------
Thank you for addressing the feedback provided. After reviewing your rebuttal and considering other reviewers' comments, I have decided to maintain my original score for your paper.

---

> ### Author Response · Authors · 2023-11-15
> **Response to Reviewer h1Dy (1/2)**
>
> Thank you for your thoughtful review and valuable feedback. In what follows, we provide point-by-point responses to your comments on the weakness of our paper and answer your questions on our paper.
>
> ---
>
> **Q1. The paper seems to heavily rely on previous works [1,2] for its theoretical results. Could you clarify the specific novel contributions of your theoretical analysis beyond the foundations laid by references [1,2]?**
>
> **Reply:** Our theoretical results significantly differ from those in [1,2]. Recall that Oono & Suzuki [1] and Cai & Wang [2] focus on showing that over-smoothing is inevitable for GCNs with ReLU or Leaky ReLU activation function. At the same time, Cai & Wang also point out that studying the normalized smoothness is an open and important problem. In contrast, our theory is novel in the following sense: 1) In Section 3, we establish ***a geometric characterization*** of how ReLU and Leaky ReLU affect the smoothness of node features, showing that ***the smoothness of node features are controllable*** by adjusting the projection of node features onto the eigenspace $\mathcal{M}$ for the first time. Our theory also informs a practical smoothness control term. 2) In Section 4, we show that adjusting the projection of node features onto the eigenspace $\mathcal{M}$ results in ***disparate effects*** on the normalized and unnormalized smoothness. 3) We study an open problem proposed by Cai & Wang. These novelties have been listed in Section 1.1 of our paper.
>
> ---
>
> **Q2. The largest non-homophily graph used is the Squirrel dataset, which consists of 5201 nodes. Exploring larger and more varied graphs could provide more comprehensive insights. Have you considered testing your approach on a broader variety of datasets, especially larger and more complex non-homophily graphs beyond the Squirrel dataset?**
>
> **Reply:** The existing experiments in our paper provide a direct comparison against the baseline models by providing a comprehensive study for each task proposed by the baseline papers. The 10 datasets we studied contain a diverse number of nodes, edges, features, and classes as shown in Table 4 in the revision. These tasks sufficiently cover a broad range of applications for GNNs.
>
> To provide broader empirical results of our techniques on large non-homophily graphs, we have included results for the Roman-empire heterophilic graph in Appendix D.5 Table 13 of the revision. The Roman-empire graph has a large diameter which supports using deeper models. We utilize the architectures from Section 6.2 which are equipped with ReLU activation functions and perform hyperparameter tuning following the same procedure. For the reviewer's ease, we have also presented the results below.
>
> | Layers   	| 2             	| 4             	| 16            	| 32            	|
> |--------------|-------------------|-------------------|-------------------|-------------------|
> | GCN 	 	| 84.48 ± 0.53  	| 84.00 ± 0.71  	| 74.56 ± 0.75  	| 14.32 ± 1.02  	|
> | GCN-SCT  	| 85.37 ± 0.56  	| 84.08 ± 0.71  	| 82.58 ± 0.57  	| 79.6 ± 0.49   	|
> | GCNII 	| 83.49 ± 0.36  	| 83.43 ± 0.40  	| 80.01 ± 0.50  	| 76.52 ± 0.70  	|
> | GCNII-SCT	| 85.44 ± 0.56  	| 85.08 ± 0.24  	| 81.44 ± 0.35  	| 77.28 ± 0.55  	|
>
> ---
>
> **Q3. The proposed method, as represented by Equation 6, appears to be a somewhat incremental modification, seemingly adding only a bias term to the graph layer. Could you elaborate on the novelty and significance of the modification introduced in Equation 6? How does the addition of a bias term fundamentally impact the model's behavior or performance?**
>
> **Reply:** The proposed smoothness control term in Equation 6 is simple but novel and different from existing works. Our innovation stems from a theoretical analysis of how ReLU and Leaky ReLU affect the smoothness of node features learned by GCN. Below we elaborate on the novelty and significance of the modification introduced in Equation 6.
>
> In Section 3, we establish a geometric characterization of how ReLU and Leaky ReLU affect the smoothness of node features learned by GCN. Our analysis demonstrates that the smoothness of node features can be modulated by adjusting the projection of node features in the eigenspace $\mathcal{M}$ -- corresponding to the largest eigenvalue of the message passing matrix $G$ in GCL. In Section 4, we provide a detailed study on how adjusting the projection of node features onto eigenspace $\mathcal{M}$ affects the smoothness of node features. Based on our theoretical analysis, we propose the smoothness control term in Equation 6. As far as we are aware, this is ***the first biased term, with guarantees to control the smoothness of the learned node features in GCL with non-linear activations***.
>
> The proposed smoothness control term effectively controls the smoothness of node features learned by GCN, resulting in feature vectors with a desired smoothness that empirically enhances node classification accuracy in GCNs.
>
> ---

---

> ### Author Response · Authors · 2023-11-15
> **Response to Reviewer h1Dy (2/2)**
>
> **Q4. The presentation of results in Table 1 could be improved for clarity and comprehensiveness.**
>
> **Reply:** Thank you for your suggestion, and we have modified the Table 1 in the revision to make it more clear and comprehensive.
>
> ---
>
> **Q5. It seems that the benefits of the proposed method diminish in deeper neural networks. Could you provide more insights into why this might be the case and whether there are ways to mitigate this limitation?**
>
> **Reply:** GCNII is a state-of-the-art GCN-style model and EGNN can be considered as GCNII with constraints on the Dirichlet energy of the node features. The proposed smoothness control term (SCT) can consistently improve the state-of-the-art baseline models, especially on heterophilic graphs. For homophilic graphs, the improvement is more significant on larger-scale datasets -- Coauthor-Physics and Ogbn-arxiv -- than on the smaller-scale datasets -- Cora, Citeseer, and PubMed. In particular, Table 1 in our paper shows that SCT can often improve the accuracy of GCNII and EGNN by more than 1% on Coauthor-Physics and Ogbn-arxiv. As far as we are aware, the performance of GCNII-SCT and EGNN-SCT is the state-of-the-art accuracy for GCN-style models; see Section 5.1 and Appendix A.2 in reference [1] listed below for details.
>
> In Table 1, though the accuracy gain is not huge in absolute values for GCNII and EGNN in some tasks, they are significant in almost all cases. To show the statistical significance of the improvement using SCT for smaller-scale datasets, we have included statistical significance tests for accuracy improvement for Cora, PubMed, and Citeseer. In particular, we train each model with 100 different random initializations using the optimal hyperparameters; the mean and standard deviation of the test accuracy are provided in Table 8 of the revision. Moreover, the t-test results on the statistical significance of the accuracy improvement are provided in Table 9 of the revision.
>
> [1] Zhou et al. Dirichlet Energy Constrained Learning for Deep Graph Neural Network, NeurIPS 2021.
>
> ---
>
> We have updated our submission based on the reviewer's feedback, with the revision highlighted in blue. We are happy to address further questions on our paper. Thank you for considering our rebuttal.

---

### Official Review · Reviewer_MfPc · 2023-10-31

**Soundness:** 3 good
**Presentation:** 2 fair
**Contribution:** 3 good
**Rating:** 6
**Confidence:** 4

**Summary:**

This paper explores how ReLU and leaky ReLU activation functions affect the smoothness of node features in Graph Convolutional Networks (GCNs). It introduces a theoretical framework and a practical algorithm to control feature smoothness. The paper's main contributions include demonstrating that these activations smooth input features without considering magnitude and proposing a learnable smoothness control term (SCT) to enhance node classification in GCNs. This work is the first to comprehensively investigate these aspects, offering insights and practical improvements for graph node classification with GCNs.

**Strengths:**

1) This paper comprehensively investigates the impact of ReLU and leaky ReLU for the first time in graph convolution, which is very meaningful.
2) The theoretical and empirical evidence presented in this paper appears to be quite sound.
3) The proposed learnable smoothness control term (SCT) can enhance the performance of existing GNN models in node classification.

**Weaknesses:**

1) The writing of this paper needs further improvement, as the theoretical part is not very easy to understand. It is recommended to add a summary of notations and optimize the formatting.
2) Experiments show that the performance improvement of SCT on deep models like GCNII and EGNN is relatively marginal.

**Questions:**

1) Please refer to the aforementioned weaknesses.
2) I don't have major concerns about this paper. My concern lies in the further improvement in writing is needed. Additionally, I haven't thoroughly reviewed the paper's proofs, and I will consider the opinions of other reviewers and relevant discussions before making a final decision.

---

> ### Author Response · Authors · 2023-11-15
> **Response to Reviewer  MfPc**
>
> Thank you for your thoughtful review, valuable feedback, and endorsement. In what follows, we provide point-by-point responses to your comments on the weakness of our paper and answer your questions on our paper.
>
>
> ---
>
> **Q1. The writing of this paper needs further improvement, as the theoretical part is not very easy to understand. It is recommended to add a summary of notations and optimize the formatting.**
>
> **Reply:** We appreciate the reviewer’s suggestion. In Appendix A of the revision, we have added a table to summarize all notations -- we did not put the table in the main text due to page limit while Section 1.3 briefly overviews the notations used in our paper.
>
> To make the theoretical part more straightforward to understand, we have further polished Section 1.1 to make it a better navigation of our theoretical results. In particular, our theoretical results are twofold: 1) In Section 3, we establish a geometric characterization of how ReLU or Leaky ReLU affects the smoothness of the node features learned by GCN, showing that the smoothness of node features are controllable by adjusting the projection of input onto the eigenspace $\mathcal{M}$. 2) In Section 4, we comprehensively examine how adjusting the projection of input to $\mathcal{M}$ affects the smoothness of node features measured by two existing smoothness notions.
>
> We have also gone through the paper and done our best to improve the formatting. We are happy to include any further suggestions from the reviewer to improve the paper.
>
>
> ---
>
> **Q2. Experiments show that the performance improvement of SCT on deep models like GCNII and EGNN is relatively marginal.**
>
> **Reply:** GCNII is a state-of-the-art GCN-style model and EGNN can be considered as GCNII with constraints on the Dirichlet energy of the node features. The proposed smoothness control term (SCT) can consistently improve the state-of-the-art baseline models, especially on heterophilic graphs. For homophilic graphs, the improvement is more significant on larger-scale datasets -- Coauthor-Physics and Ogbn-arxiv -- than on the smaller-scale datasets -- Cora, Citeseer, and PubMed. In particular, Table 1 in our paper shows that SCT can often improve the accuracy of GCNII and EGNN by more than 1% on Coauthor-Physics and Ogbn-arxiv. As far as we are aware, the performance of GCNII-SCT and EGNN-SCT is the state-of-the-art accuracy for GCN-style models; see Section 5.1 and Appendix A.2 in reference [1] listed below for details.
>
> In Table 1, though the accuracy gain is not huge in absolute values for GCNII and EGNN in some tasks, they are significant in almost all cases. To show the statistical significance of the improvement using SCT for smaller-scale datasets, we have included statistical significance tests for accuracy improvement for Cora, PubMed, and Citeseer. In particular, we train each model with 100 different random initializations using the optimal hyperparameters; the mean and standard deviation of the test accuracy are provided in Table 8 of the revision. Moreover, the t-test results on the statistical significance of the accuracy improvement are provided in Table 9 of the revision.
>
> [1] Zhou et al. Dirichlet Energy Constrained Learning for Deep Graph Neural Network, NeurIPS 2021.
>
>
> ---
>
> **Q3. I don't have major concerns about this paper. My concern lies in the further improvement in writing is needed.**
>
>
> **Reply:** Thank you for your comment. We have further improved the writing in the revision to make the paper clearer, and we are happy to include any further suggestions from the reviewer.
>
> ---
>
> We have updated our submission based on the reviewer's feedback, with the revision highlighted in blue. We are happy to address further questions on our paper. Thank you for considering our rebuttal.

---

> > ### Comment · Reviewer_MfPc · 2023-11-22
> > **Re**
> >
> > Thank you for the author's response. I have already read it. The author's answer has addressed my concerns, and I will keep my current score and discuss it with other reviewers.

---

> > > ### Author Response · Authors · 2023-11-22
> > > **Further Response to Reveiwer MfPc**
> > >
> > > Dear Reviewer MfPc,
> > >
> > > Thank you for considering our rebuttal. We are open to addressing any further questions or concerns on our paper.

---

### Official Review · Reviewer_Pp4G · 2023-10-31

**Soundness:** 3 good
**Presentation:** 3 good
**Contribution:** 3 good
**Rating:** 6
**Confidence:** 3

**Summary:**

This paper first shows that the image of ReLU and Leaky ReLU is contained in a particular sphere. As a corollary, alternative proofs are given to the contraction property of ReLU and Leaky ReLU (with respect to the distance to the eigenspace $\mathcal{M}$). Then, this paper defines the normalized smooth index $s(\cdot)$ for each feature dimension and elucidate how the parallel component of a feature vector to $\mathcal{M}$ affects the change of $s(\cdot)$ by applying ReLU and Leaky ReLU. Based on this, this paper proposes Smoothness Control Term (SCT) to adjust the feature component parallel to $\mathcal{M}$ as bias terms of GNN layers. SCT is applied to GCNII and EGCN models and evaluates its performance on five Citation Network datasets and node classification problems on five heterophilic datasets.

**Strengths:**

- The proposed method improves prediction accuracy, especially for datasets with high heterophily (Table 2). This result is consistent with the claim that the proposed method is effective for over-smoothing.
- The proposed method applies to most GNNs of MPNN type, although numerical verifications are limited to GCNII and ECGN.
- The proof is carefully written and easy to follow.

**Weaknesses:**

- The proof about the contraction property applies only to ReLU and Leaky ReLU. Therefore, this theoretical analysis does not broaden the applicable GNN types.

**Questions:**

* P.2: *We prove that there is a high-dimensional sphere ... ReLU or leaky ReLU*: It is difficult to grasp what is intended by this sentence alone. It would be better to be more specific. For example, *We prove the output of ReLU or Leaky ReLU lies in a high-dimensional space characterized by the input.*
* P.5, Definition 4.1: $\|\boldsymbol{z}_{\mathcal{M}}^{(i)}\|$ is undefined.
* P.7: If I understand correctly, the $\beta_l$ parametrization comes from the work of GCNII. If this paper references it, the paper should be cited explicitly.
* P.7: $\boldsymbol{W}^1$ -> $\boldsymbol{W}^l$
* P.8, Table 1: The column "16 Layers" is not aligned.

**Details Of Ethics Concerns:**

N.A.

---

> ### Author Response · Authors · 2023-11-15
> **Response to Reviewer Pp4G**
>
> Thank you for your thoughtful review and valuable feedback. In what follows, we provide point-by-point responses to your comments on the weakness of our paper and answer your questions on our paper.
>
> ---
>
> **Q1. P.2: We prove that there is a high-dimensional sphere ... ReLU or Leaky ReLU: It is difficult to grasp what is intended by this sentence alone. It would be better to be more specific. For example, We prove the output of ReLU or Leaky ReLU lies in a high-dimensional space characterized by the input.**
>
>
> **Reply:** Thank you for your suggestion. To be more specific, we have updated the statement as follows: We prove that the projection of the output of ReLU/Leaky ReLU onto the eigenspace $\mathcal{M}^\perp$ -- corresponding to eigenvalue 1 of matrix ${\bf G}$ in equation (1) -- lies in a high-dimensional sphere, whose center only depends on the input but the radius depends on both input and output of ReLU/Leaky ReLU.
>
> ---
>
> **Q2. P.5, Definition 4.1: $||\mathbf{z}_{\mathcal{M}}^{(i)}||$ is undefined.**
>
>
> **Reply:** We have defined $||\mathbf{z}_{\mathcal{M}}^{(i)}||$ in Definition 4.1 in the revised paper.
>
> ---
>
> **Q3. P.7: If I understand correctly, the $\beta_l$ parametrization comes from the work of GCNII. If this paper references it, the paper should be cited explicitly.**
>
>
> **Reply:** We have cited the GCNII paper right after the $\beta_l$ parameterization in the revision.
>
> ---
>
> **Q4. P.7: $\mathbf{W}^1 \rightarrow \mathbf{W}^l$**
>
> **Reply:** Thank you for your very careful review. This is not a typo, and this is precisely what EGNN uses. In our revision, we refer to the orthogonal initialization and orthogonal regularization in the original EGNN paper, i.e., reference [42]. In summary, EGNN initializes the first weight matrix $\mathbf{W}^1$ as a diagonal matrix $\sqrt{c_{\max}}\cdot \mathbf{I}$, and the subsequent weight matrices $\mathbf{W}^l$ for $l>1$ as diagonal matrices $\mathbf{I}$. Subsequently, orthogonal regularization is applied to penalize the distances between the trainable weights $\mathbf{W}^1$ and $\mathbf{W}^l$ and the initial weights.
>
> ---
>
> **Q5. P.8, Table 1: The column "16 Layers" is not aligned.**
>
> **Reply:** This has been fixed in the revised version.
>
> **Q6. The proof about the contraction property applies only to ReLU and Leaky ReLU. Therefore, this theoretical analysis does not broaden the applicable GNN types.**
>
>
> **Reply:** Understanding whether over-smoothing happens or not and how to control the smoothness when other activation functions are used is an interesting problem. However, the over-smoothing has only been theoretically justified for GCN using ReLU and Leaky ReLU activation functions.
>
> To be more specific, the established proofs of over-smoothing by Oono & Suzuki (ICLR 2020) and Cai & Wang (arXiv:2006.13318) rely on the contraction property of ReLU and Leaky ReLU. In particular, Oono & Suzuki (ICLR 2020) show over-smoothing for GCN with ReLU by using the contraction property of ReLU and point out that it is hard even to extend this result to Leaky ReLU. Cai & Wang prove the contraction property of ReLU or Leaky ReLU by using Dirichlet energy to characterize the smoothness of node features. Importantly, they highlight that other activation functions, such as Sigmoid and Tanh, may not exhibit the contraction property.
>
> Nevertheless, our work builds a geometric understanding of the contract property of ReLU and Leaky ReLU, which further informs a practical approach to control the smoothness of the learned node features by GCN with ReLU or Leaky ReLU activation function.
>
> ---
>
> We have updated our submission based on the reviewer's feedback, with the revision highlighted in blue. We are happy to address further questions on our paper. Thank you for considering our rebuttal.

---

> > ### Comment · Reviewer_Pp4G · 2023-11-22
> > **Reponses to authors' comments**
> >
> > I thank the authors for answering my review comments. Here, I respond to the authors' comments one by one.
> >
> > Q.1: OK
> >
> > Q.2: OK
> >
> > Q.3: OK
> >
> > Q.4: OK. Thank you for your detailed explanation and I am sorry for my misunderstanding.
> >
> > Q.5: OK
> >
> > Q.6:
> >
> > > However, the over-smoothing has only been theoretically justified for GCN using ReLU and Leaky ReLU activation functions.
> >
> > Thank you for the explanation. My point was that it would be a plus if either:
> > 1. the theory can explain the over-smoothing for activation functions that are practically observed but have not been theoretically justified or
> > 2. the theory can predict the over-smoothing for (possibly new) activation functions other than ReLU and Leaky ReLU and confirm the phenomena practically.

---

> ### Author Response · Authors · 2023-11-22
> **Further Response to Reviewer Pp4G**
>
> Dear Reviewer Pp4G,
>
> Thank you for considering our rebuttal and for your invaluable feedback. The two problems you pointed out are both very interesting and thank you again.
>
> Our study was motivated by an empirical finding by Oono & Suzuki and the open problem pointed out by Cai & Wang rather than justifying over-smoothing for GCN with ReLU or leaky ReLU again. In particular, Oono & Suzuki empirically find that the accuracy of GCN is strongly correlated with a normalized smoothness-related quantity, which motivates us to design practical algorithms to let GCN automatically learn the desired smoothness of node features to improve classification accuracy. Moreover, Cai & Wang pointed out that studying the over-smoothing measured by normalized smoothness is an important open problem.
>
> To theoretically study the empirical findings by Oono & Suzuki and the open problem pointed out by Cai & Wang, we first establish a geometric relation between the input and output of ReLU and leaky ReLU. Our established geometric relation not only confirms over-smoothing but also informs a new smoothness control term that can let GCN learn a desired smoothness to improve the classification accuracy. Our established geometric relation leverages the special structure of ReLU and leaky ReLU. It is possible but highly nontrivial to establish a similar geometric relation for other activation functions.
>
> We further study the disparate effects of our proposed smoothness control term on the normalized and unnormalized smoothness of node features. Our theoretical study shows that each GCN layer -- even with our proposed smoothness control term -- smooths node features when measured by the unnormalized smoothness. In contrast, the GCN layer with our proposed smoothness control term can increase, decrease, and preserve the smoothness of node features when measured by the normalized smoothness.
>
> Moreover, our empirical results confirm that the proposed smoothness control term can effectively improve the classification accuracy of GCN and related models.
>
> Thank you for considering our rebuttal. We are open to addressing any further questions or concerns on our paper.

---

### Author Response · Authors · 2023-11-15
**General Response (1/2)**

Dear reviewers and AC,

We thank the reviewers for their thoughtful reviews and valuable feedback, which have helped us significantly improve the paper. We appreciate the reviewers’ acknowledgment of the strengths of our paper. In this general response, we would like to address common comments, express our concerns about reviewer pkn8’s unethical review, and summarize our revisions.

-----

**Contributions and novelty over the work**

The two pioneering papers Oono & Suzuki (ICLR 2020) and Cai & Wang (arXiv:2006.13318) prove that over-smoothing happens for GCN with ReLU and Leaky ReLU activation functions by studying the distance of node features to the eigenspace $\mathcal{M}$ and the Dirichlet energy of node features.

Compared to the two pioneering papers, the key contribution and novelty of our work are threefold: (1) We establish a new geometric characterization of how ReLU and Leaky ReLU affect the smoothness of the node features learned by GCN, which proves that the smoothness of node features is controllable by adjusting the projection of node features onto the eigenspace $\mathcal{M}$ for the first time. (2) Informed by our geometric characterization, we comprehensively study how adjusting the projection of node features onto $\mathcal{M}$ affects the smoothness of node features measured by both unnormalized and normalized smoothness notions -- the only two existing notions used by the community. A closely related concept to normalized smoothness has been first empirically studied by Oono & Suzuki and Cai & Wang point out that studying the normalized
smoothness is an important open problem. (3) Based on our theory, we propose a smoothness control term to let GCN-style models learn node features with a desired smoothness to improve the node classification.

-----

**The significance of performance improvement on GCNII and EGNN**

GCNII is a state-of-the-art GCN-style model and EGNN can be considered as GCNII with constraints on the Dirichlet energy of the node features. The proposed smoothness control term (SCT) can consistently improve the state-of-the-art baseline models, especially on heterophilic graphs. For homophilic graphs, the improvement is more significant on larger-scale datasets -- Coauthor-Physics and Ogbn-arxiv -- than on the smaller-scale datasets -- Cora, Citeseer, and PubMed. In particular, Table 1 in our paper shows that SCT can often improve the accuracy of GCNII and EGNN by more than 1% on Coauthor-Physics and Ogbn-arxiv. As far as we are aware, the performance of GCNII-SCT and EGNN-SCT is the state-of-the-art accuracy for GCN-style models; see Section 5.1 and Appendix A.2 in reference [1] listed below for details.

In Table 1, though the accuracy gain is not huge in absolute values for GCNII and EGNN in some tasks, they are significant in almost all cases. To show the statistical significance of the improvement using SCT for smaller-scale datasets, we have included statistical significance tests for accuracy improvement for Cora, PubMed, and Citeseer. In particular, we train each model with 100 different random initializations using the optimal hyperparameters; the mean and standard deviation of the test accuracy are provided in Table 8 of the revision. Moreover, the t-test results on the statistical significance of the accuracy improvement are provided in Table 9 of the revision.

[1] Zhou et al. Dirichlet Energy Constrained Learning for Deep Graph Neural Network, NeurIPS 2021.

-----

---

> ### Author Response · Authors · 2023-11-15
> **General Response (2/2)**
>
> **Reviewer pkn8’s unethical and insulting comments about normalized smoothness is meaningless to study**
>
> ***The criticism that the normalized smoothness is meaningless to study is rootless, unethical, and insulting. Moreover, the reviewer falsifies the truth.*** Criticizing another's paper by falsifying the truth is scientific misconduct. It is a serious offense that can damage the reputation of other scientists and undermine public trust in science.
>
> ***Empirical evidence for studying normalized smoothness:*** The correlation between the relative magnitude $||{\bf z}\_{\mathcal{M}^\perp}||/||{\bf z}\_\mathcal{M}||$ -- a quantity that is closely related to the normalized smoothness -- and classification accuracy was first empirically studied in Figure 4 by Oono & Suzuki (ICLR 2020).
>
> ***Theoretical evidence for studying normalized smoothness:*** Studying normalized smoothness is an open important problem pointed out by Cai & Wang (arXiv:2006.13318). We believe that studying an important open problem, pointed out in the pioneering paper, as part of our paper is not a weakness.
>
> We respectfully but strongly disagree with reviewer pkn8’s criticism that the existing normalized smoothness notion disconnects node smoothness from model performance, making it carry no insights into practice and thus it is meaningless to study  -- we believe this comment is highly questionable, counter to the empirical study by Oono & Suzuki, and irrelevant to GCN. ***Reviewer pkn8’s unethical comments come from falsifying the truth.***
>
> First, ***GCN vs. linear classifier:*** Reviewer pkn8 raises an example to criticize that the normalized smoothness is not valid for interpreting the classification results of a linear classifier because the bias term in the linear classifier can accommodate the shift in node features. However, ***we are studying GCN rather than linear classifier***. The normalized smoothness notion is used to characterize the smoothness of node features learned by GCN rather than the linear classifier. Moreover, the benchmark GCN model directly passes the node features to log\_softmax to get the classification results rather than using a linear classifier; see https://github.com/tkipf/pygcn/blob/master/pygcn/models.py.
>
> Second, ***training vs. testing of linear classifier:*** Though irrelevant to our work, we would like to comment on reviewer pkn8’s comment that normalized smoothness lacks interpretability of the classification results of the linear classifier, say $softmax(Wx+b)$. If the features of the training set are shifted, the bias $b$ can accommodate it during training. However, notice that the linear classifier is first trained and then applied to new data. Once the model is trained, we will have a fixed set of weights $W$ and $b$. For fixed $W$ and $b$, $softmax(Wx+b)$ changes when $x$ is shifted.
>
> Reviewer pkn8 also ***falsifies that there is no empirical or theoretical evidence*** for the statement that “[5] points out that over-smoothing -- measured by the distance of node features to the eigenspace $\mathcal{M}$ or the Dirichlet energy -- is a misnomer, and the real smoothness of a graph signal should be characterized by a normalized smoothness, e.g., normalizing the Dirichlet energy by the magnitude of the features.” ***This is just a rephrase of the statement in Cai & Wang’s paper.*** The empirical evidence has been provided in Oono & Suzuki’s paper.
>
> -----
>
> **Summary of the major revision**
>
> Incorporating the comments and suggestions from all reviewers, besides fixing typos and reformating the paper, we have made the following major changes in the revision:
>
> - Adding additional experiments to show the statistical significance of the accuracy improvement by using the proposed smoothness control term; see Appendix D.3, Tables 8 and 9 for details.
>
> - Conducting some new experiments on larger-scale heterophilic graph node classification tasks, namely Roman-empire, to further confirm the effectiveness of the proposed smoothness control term; see Appendix D.5 and Table 13 in the revision for details.
>
> -----
>
> Thank you for considering our rebuttal and the concerns about the unethical review by reviewer pkn8.

---

### Author Response · Authors · 2023-11-20
**Seeking Feedback and Further Guidance**

Dear Reviewers,

As we approach the end of the discussion phase, we note that we have yet to receive any responses from your end. We fully appreciate that thoughtful reviewing requires time and effort, and we respect this aspect of the process.


However, we are eager to understand if there are any further concerns or areas for improvement in our submission that we can address. Your specialized insights and feedback are invaluable to the development of our work. If you could take the time to review our rebuttal and provide any additional comments, it would be greatly appreciated.


Thank you very much for your time and dedication to this process. We look forward to your valuable feedback.


Best regards,


Authors

---

### Meta-Review · Area_Chair_kAH7 · 2023-12-09

**Metareview:**

“Smoothing” is the phenomena by which node features become homogeneous after a graph convolutional network (GCN) layer. Authors study the “over-smoothing” hypothesis that proposes that smoothing of features explain the decrease performance of GCN with increase in number of layers. Specifically, they study the notion of “normalized smoothness” in GCNs without attention, and this notion was hinted in previous works. They show that contrary to previous notions of smoothness, GCN layers with ReLU or LeakyReLU activation can decrease or preserve the normalized smoothness as opposed to just increasing the smoothness. Their analysis also enables re-derivation of effect of these layers on prior notions of smoothness.

Inspired from their analysis, author propose and experiment with a new class of GCN layers (called SCT) which can learn to increase or decrease normalized or un-normalized notions of smoothness.

Even though the experimental results are promising (especially significant results on Coauther-Physics dataset), there remains some questions and concerns.

For example, normalized smoothness is never quantified for the new architecture and baselines. It isn’t fully clear where the architecture came from. “The proposed SCT can change the smoothness, in terms of both normalized and unnormalized smoothness, of the learned node features by GCN”: Where is this claim supported? It is also not shown whether SCT enables training deeper models. This points to a disconnect between the theory and experiments sections.

How do EGNN have more parameters than EGNN-SCT? Doesn’t SCT add more parameters? Why is the SCT not compared with other prior work aiming to tackle oversmoothing? How the current baselines were chosen?

Why is validation metric reported instead of test metric (as far as I understand)? Further, it is not clear whether Bayesian Optimization for their method helped with the improved performance. It would have been better if baselines were also optimized using Bayesian Optimization. Or are they? Do the proposed methods using a larger grid than baselines for hyperparameter search?

Big improvements in heterophilic datasets needs to be studied more. Further it is not clear why EGNN was eliminated from this comparison on heterophilic datasets.

**Justification For Why Not Higher Score:**

Unanswered questions around experiments.

**Justification For Why Not Lower Score:**

N/A

---

### Decision · Program_Chairs · 2024-01-16

Reject